# GEM: Empowering MLLM for Grounded ECG Understanding with Time Series and Images

**Xiang Lan**[1], **Feng Wu**[1], **Kai He**[1], **Qinghao Zhao**[2], **Shenda Hong**[3✉], **Mengling Feng**[1✉]

[1]National University of Singapore [2]Peking University People's Hospital [3]Peking University

✉Correspondence to: `hongshenda@pku.edu.cn`, `mornin@nus.edu.sg`

## Abstract

While recent multimodal large language models (MLLMs) have advanced automated ECG interpretation, they still face two key limitations: (1) insufficient multimodal synergy between ECG time series and ECG images, and (2) limited explainability in linking diagnoses to granular waveform evidence. We introduce GEM, the first MLLM unifying ECG time series, 12-lead ECG images and text for grounded and clinician-aligned ECG interpretation. GEM enables feature-grounded analysis, evidence-driven reasoning, and a clinician-like diagnostic process through three core innovations: a dual-encoder framework extracting complementary time series and image features, cross-modal alignment for effective multimodal understanding, and knowledge-guided instruction data generation for generating high-granularity grounding data (ECG-Grounding) linking diagnoses to measurable parameters (*e.g.*, QRS/PR Intervals). Additionally, we propose the Grounded ECG Understanding task, a clinically motivated benchmark designed to comprehensively assess the MLLM's capability in grounded ECG understanding. Experimental results on both existing and our proposed benchmarks show GEM significantly improves predictive performance (CSN $7.4\%\uparrow$), explainability ($22.7\%\uparrow$), and grounding ($25.3\%\uparrow$), making it a promising approach for real-world clinical applications. Codes, model, and data are available at https://github.com/lanxiang1017/GEM.

## 1 Introduction

Electrocardiography (ECG), a cornerstone of cardiac diagnostics, captures the heart's electrical activity through body-surface electrodes, enabling non-invasive assessment of cardiac physiology and pathology [Berkaya et al., 2018, Hannun et al., 2019]. Clinical ECG interpretation synergizes computational and clinical expertise: automated algorithms process raw signals to detect fiducial points (*e.g.*, R-wave peaks) and generate diagnostic hypotheses, while clinicians validate these findings through 12-lead waveform analysis [Smulyan, 2019]. By contextualizing algorithmic outputs with patient-specific factors, clinicians resolve ambiguities, detect subtle anomalies, and formulate diagnoses. This synergy between computational precision and expert judgment ensures reliable and holistic diagnoses in clinical practice.

Deep learning models have achieved promising results in cardiac anomalies detection [Hong et al., 2020, Zhu et al., 2021, Kiyasseh et al., 2021, Lan et al., 2022, Yang et al., 2023, Zhao et al., 2024a, Li et al., 2024, Lan et al., 2024, McKeen et al., 2024] yet lack language capability and model explainability. While recent MLLMs like PULSE [Liu et al., 2024b] have advanced language-based ECG interpretation via large-scale instruction tuning (*e.g.*, ECG-Instruct's 1M+ samples), they primarily focus on static image inputs and predefined diagnostic tasks, leaving two critical gaps: **Limited Modality Exploration**. Current models typically process only one non-text modality, overlooking the benefits of a synergistic approach that combining time series and image-based modeling. For example, time series models capture dynamic changes but may overlook spatial

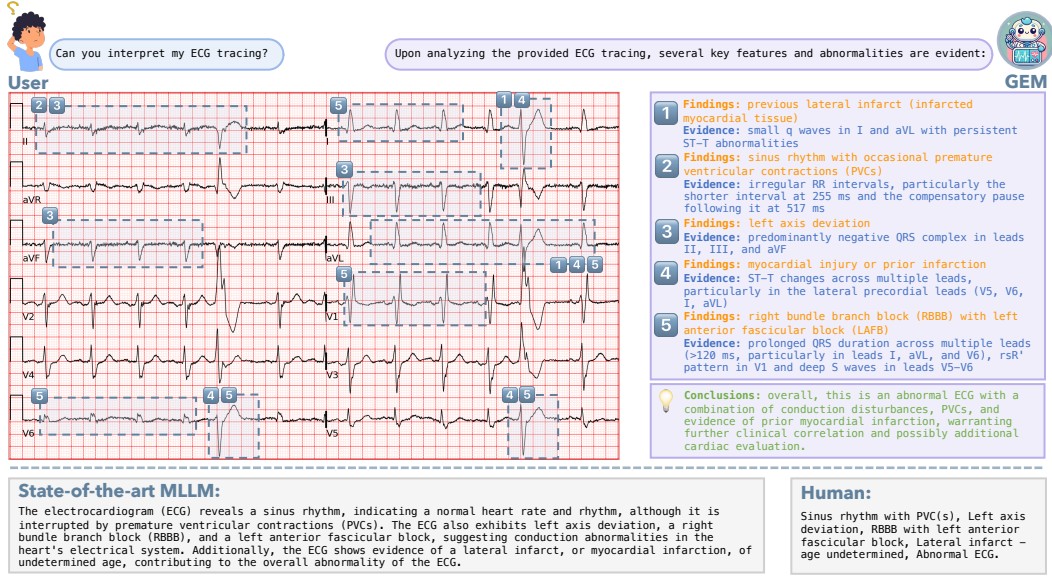

Figure 1: GEM offers superior granularity in ECG interpretation compared to state-of-the-art models and human-written reports.

patterns, while image-based models detect global structures but may miss subtle temporal details. This limits their ability to replicate clinicians' holistic reasoning, which integrates both machine-measured temporal signals and waveform patterns from 12-lead plots. **Insufficient Explainability and Grounding**. Most existing models designed for ECG provides limited explainability, failing to explicitly connect diagnoses to granular waveform evidence and provide insight into their diagnostic reasoning. A trustworthy ECG model should not only predict cardiac conditions but also explicitly highlight which ECG features led to those conclusions. Such grounded explanations enhance transparency and make model outputs more reliable and actionable for clinical decision-making.

In this work, we introduce GEM, a multimodal large language model designed for grounded ECG understanding by integrating time series, image, and text data. As a conversational cardiology AI assistant, GEM differs from other MLLMs through three key features. First, it provides *feature-grounded analysis*, ensuring that its findings are explicitly tied to detailed ECG features. Second, it offers *evidence-driven diagnosis*, where its conclusions are supported by clear and logical reasoning directly linked to ECG findings. Lastly, GEM simulates a *realistic interpretation process*, mimicking how a clinician analyzes ECGs and arrive at a diagnosis.

Achieving these capabilities entails challenges in two key dimensions. On the modeling side, it is crucial to effectively integrate information from different modalities to support accurate ECG understanding. On the data side, there is currently no available instruction data designed for training LLMs on high-granularity ECG interpretation.

We tackle these challenges through three novel approaches. *Multimodal Encoding* allows GEM to extract and integrate complementary features from raw ECG time series and their transformed images. It employs a dual-encoder architecture, with each encoder specialized in its respective modality using established models from the time series and vision domains. This design leverages the unique strengths of both modalities. *Cross-modal Alignment Learning* facilitates the interpretation of multimodal ECG data by the LLM. Time series representations are first projected to image representations dimensionality, followed by a shared projector that transforms both into language-like embeddings that are comprehensible to LLM. These aligned embeddings are then fused with textual instruction embeddings, enabling effective multimodal understanding through next-token prediction training. *Knowledge-guided Instruction Data Generation* supports the construction of high-granularity instruction data annotated with heartbeat-level physiological features, without the need for manual annotation. This methodology integrates a grounding feature extractor, which derives precise physiological features from ECG time series, and a cardiology-specific diagnosis guider, which processes these features into structured prompts to more effectively leverage GPT-4o's

latent medical knowledge for generating clinically detailed and feature-grounded ECG instruction data. Ultimately, GEM delivers significantly more detailed and informative interpretations than both human-written reports and leading MLLMs, as shown in Figure 1.

The main contributions of this work are three-folds:

*1.First Unified Multimodal ECG Model.* We present GEM, the first multimodal framework to synergistically integrate raw ECG time seriesa, 12-lead ECG plots, and textual instructions, leveraging their complementary strengths to advance grounded ECG understanding.

*2.First High-granularity ECG Grounding Dataset.* We propose a novel knowledge-guided instruction data generation method, resulting in ECG-Grounding, a dataset comprising 30,000 instruction pairs annotated with heartbeat-level physiological features. This is the first high-granularity ECG grounding dataset, enabling evidence-based diagnosis and improving the trustworthiness of medical AI.

*3.Clinically Oriented Diagnostic System.* We introduce the Grounded ECG Understanding task, a clinically motivated benchmark designed to comprehensively assess a model's ECG interpretation capability. Experimental results demonstrate that GEM not only excels in predictive performance but also significantly enhances explainability and grounding, making it more applicable for real-world clinical settings while fostering greater trust among medical professionals.

## 2  Related Work

### 2.1  Multimodal Large Language Models

Large Language Models (LLMs), such as GPTs [Achiam et al., 2023], LLaMA [Touvron et al., 2023], have made significant advancements in artificial intelligence. Despite their superior performance on numerous natural language processing tasks, LLMs are inherently limited to the text modality, making them "blind" to other modalities such as images, audio and video. To mitigate this constraint, Multimodal Large Language Models have been recently developed to extend the ability of LLMs in comprehending multiple modalities [Liu et al., 2024a, Zhang et al., 2024, 2025, Huang et al., 2025]. By integrating LLMs with various data sources, MLLMs enable the handling of diverse information beyond text. For example, LLaVA [Liu et al., 2024a] enables LLMs to comprehend visual inputs by adopting a learnable projector to map image features into the word embedding space. Video-LLaMA [Zhang et al., 2023] further enhances LLMs by enabling video perception and understanding. Qwen-Audio [Chu et al., 2023] introduces an audio-language model capable of processing various audio types, including human speech, natural sounds, and music. Medical data, by nature, are inherently multimodal, encompassing diverse formats such as images, physiological time series, and textual reports. These modalities collectively form the foundation for clinical decision-making, driving the advancement of AI in medical applications [Li et al., 2023, Moor et al., 2023, Radhakrishnan et al., 2023, Liu et al., 2023, Hong and Hong, 2023, Zhu et al., 2024, Ren et al., 2024, Sellergren et al., 2025, Yang et al., 2025, Jin et al., 2025, He et al., 2025]. Different from these works, our focus is on empowering MLLMs with the grounded ECG understanding capability.

### 2.2  Language-based ECG Analysis

Language-based ECG diagnosis and interpretation is still in its early stages of development. Only a few recent studies have explored LLM-based approaches for ECG analysis. For instance, Yu et al. [2023] proposes a zero-shot retrieval-augmented diagnosis technique, embedding domain knowledge from textbooks and research papers into a vector database to improve zero-shot diagnostic accuracy. Cai et al. [2023] proposes JoLT, a framework that jointly models ECG time series and text using a Querying Transformer to align their representations. Liu et al. [2024b] introduces PULSE, an LLM-based framework designed to enhance ECG image understanding for diagnosis and report generation. PULSE synthesizes realistic ECG images from raw ECG signals, enabling better utilization of image-based LLaVA models. Zhao et al. [2024b] develops ECG-CoCa, an ECG encoder trained on ECG-text pairs, alongside ECG-Chat, a modified LLaVA model capable of processing ECG time series. Chan et al. [2024] proposes an analytical framework integrating time series data with LLMs, combining physiological signal analysis with contextual textual information. Wan et al. [2024] designs an instruction-tuning framework for automated ECG report generation, converting ECG-text pairs into chatbot-style instructions and fine-tuning the LLM's linear layers.

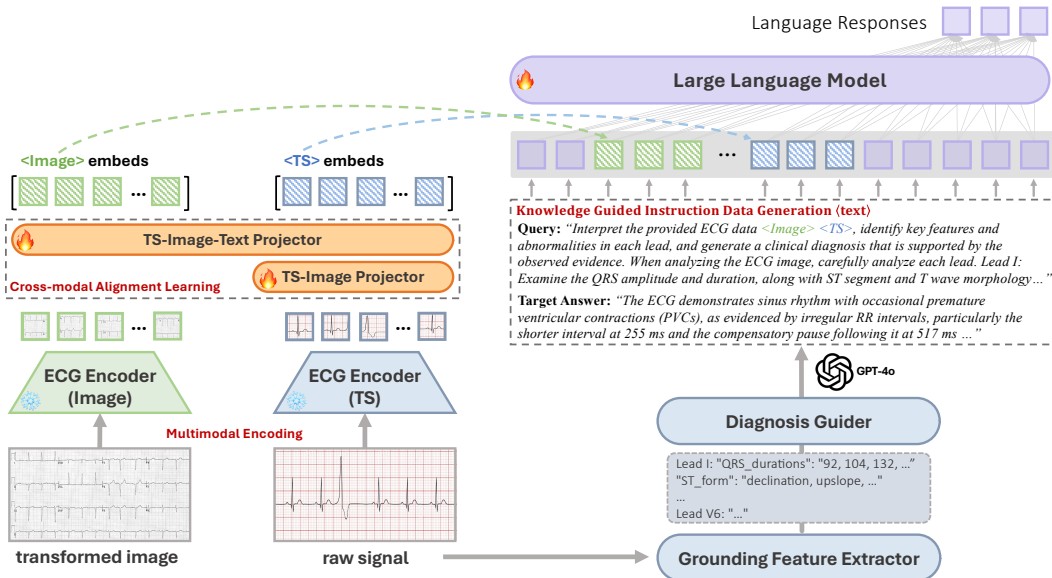

Figure 2: GEM's Architecture. *Multimodal Encoding*: Separate encoders process ECG time series and images to generate modality-specific representations, enabling a holistic analysis of ECG data. *Cross-modal Alignment Learning*: Time series and image representations are first aligned and then mapped to a textual space using a shared projector, ensuring coherent understanding for the LLM. *Knowledge-guided Instruction Data Generation*: Physiological features extracted from all 12 leads are sequenced and structured using a diagnosis guider, which prompts GPT-4o with domain-specific instructions to generate high-granularity instructional data.

GEM distinguishes itself from existing language-based ECG models in two key aspects. First, in model architecture, unlike existing methods that limit analysis to isolated modalities (either ECG signals or transformed ECG images) with text, GEM introduces a unified architecture that integrates both time series data and 12-lead images. This approach mirrors a clinician's natural workflow, where dynamic signal trends and spatial waveform patterns are jointly analyzed for a more comprehensive interpretation. Second, in ECG understanding, GEM establishes a new paradigm for evidence-driven diagnosis. While current models often lack grounded understanding, GEM enables heartbeat-level interpretability by directly linking each diagnostic conclusion to quantifiable physiological evidence, enhancing explainability and clinical reliability. By combining the two paradigm-shifting innovations, GEM addresses fundamental limitations in existing models and advances language-based ECG analysis, setting a new standard for conversational AI-assisted cardiac diagnostics.

## 3  Method

### 3.1  Overview

GEM's training primarily relies on three key components: the multimodal encoding, the cross-modal alignment learning, and the knowledge-guided instruction data generation. Figure 2 provides an overview of the GEM model. We will elaborate on each components in the following sections.

### 3.2  Multimodal Encoding

We represent the ECG time series as $\boldsymbol{x}_{ts} \in \mathbb{R}^{C \times L}$, where $C$ is the number of leads in multi-lead ECG and $L$ is the length of the signal. The transformed ECG image derived from the time series is denoted as $\boldsymbol{x}_{img} \in \mathbb{R}^{H \times W \times 3}$, where $H$ and $W$ are the height and width of the image, respectively. For ECG time series encoder $E_{ts}(\theta_{ts})$, we adopt the pre-trained ECG-CoCa model [Zhao et al., 2024b], which has been extensively trained on a large number of ECG-Text pairs with contrastive learning to effectively capture the intricate patterns within the ECG time series:

$$\mathbf{e}_{ts} \in \mathbb{R}^{n_s \times d_s} = E_{ts}(\boldsymbol{x}_{ts}|\theta_{E_{ts}}), \tag{1}$$

where $n_s$ is the number of time series feature patches and $d_s$ is the dimension of time series features. For ECG image encoder $E_{img}(\theta_{img})$, we utilize the pre-trained CLIP encoder from LLaVA [Liu et al., 2024a]. This model is adept at understanding and processing visual information, making it suitable for extracting features from the ECG images:

$$\mathbf{e}_{img} \in \mathbb{R}^{n_m \times d_m} = E_{img}(\boldsymbol{x}_{img}|\theta_{E_{img}}), \tag{2}$$

where $n_m$ is the number of image feature patches and $d_m$ is the dimension of image features. These two encoders enable the separate extraction of features from time series and images. This dual-encoder approach allows us to harness the distinct advantages of each data type, enhancing the overall interpretative power of our model.

## 3.3 Cross-modal Alignment Learning

Considering that the time series and image encoders are trained independently, their generated representations often show inconsistencies within the representation space. Meanwhile, to ensure that the LLM can interpret ECG time series and images effectively, it is essential that these multimodal inputs are rendered as comprehensible as textual data. Therefore, aligning these diverse modality inputs within a unified representation space becomes crucial.

In our approach, the ECG time series representation is first mapping to the same dimensionality as the ECG image representation. This is accomplished using a multi-layer perceptron (MLP) projector:

$$\hat{\mathbf{e}}_{ts} \in \mathbb{R}^{n_s \times d_m} = MLP_{ts}(\mathbf{e}_{ts}|\theta_{M_{ts}}). \tag{3}$$

Subsequently, we employ an additional projector to map both the time series and image representations into a consistent textual space:

$$\mathbf{h}_{ts} \in \mathbb{R}^{n_s \times d_t} = MLP(\hat{\mathbf{e}}_{ts}|\theta_M), \tag{4}$$

$$\mathbf{h}_{img} \in \mathbb{R}^{n_m \times d_t} = MLP(\mathbf{e}_{img}|\theta_M), \tag{5}$$

where $d_t$ is the dimension of the text embeddings. This step is for ensuring that the features extracted from both modalities are not only aligned dimensionally but are also interpretable in a LLM-friendly format.

Once we have obtained the features from both the time series and image modalities, we integrate these with the embeddings of the textual query $\boldsymbol{x}_q$:

$$\mathbf{x} = \text{Concatenate}(\mathbf{h}_{ts}, \mathbf{h}_{img}, \text{Embeded}(\boldsymbol{x}_q)). \tag{6}$$

The integration is crucial for creating a cohesive representation that encapsulates the full spectrum of information from the multimodal inputs.

## 3.4 Knowledge-guided Instruction Data Generation

Instruction data forms the foundation of multimodal training, directly shaping how the MLLM generates responses for given queries. This is because the language response $\theta_{LLM}(\mathbf{x})$ is optimized to match the target answer $\boldsymbol{y}$ for each multimodal input $(\boldsymbol{x}_{ts}, \boldsymbol{x}_{img}, \boldsymbol{x}_q)$. Here, we introduce two key mechanisms to guarantee that $\boldsymbol{y}$ aligns with feature-grounded analysis, evidence-driven diagnosis, and a realistic interpretation process, thereby empowering GEM with a grounded understanding of ECG data.

**Grounding Feature Extractor.** To enable the feature-grounded analysis, we propose to further excavate more detailed information from raw ECG time series. We begin by extracting universal elements, including waveforms, amplitudes of fiducial points, and intervals, from each heartbeat in each lead. These elements are then structured into a time-ordered sequence for further analysis. For instance, an ECG time series with ten visible heartbeats allows us to construct a QRS duration sequence as $[QRS_1, QRS_2, ..., QRS_{10}]$, where each element precisely represents the QRS duration for a specific heartbeat. These sequences provide a fine grade description of the physiological features of the ECG, enabling the model to assess heart conditions at the level of individual heartbeats. In our implementation, we incorporate 14 feature sequences for each of the 12 ECG leads, covering: Heart Rate, RR Interval 1, RR Interval 2, P Amplitude, P Duration, PR Interval, QRS Amplitude, QRS Duration, T Amplitude, T Duration, ST Duration, ST Form, QT Interval, and QTc Interval.

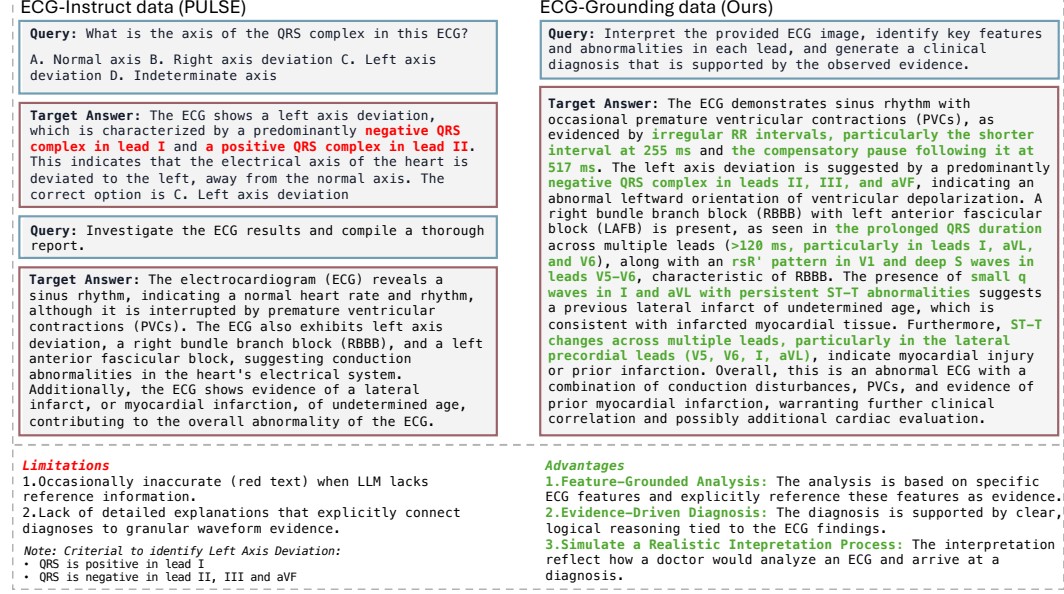

Figure 3: Comparison of ECG-Instruct and our ECG-Grounding.

This comprehensive set of sequences captures the temporal evolution of key physiological features, enabling granular analysis of cardiac activity. The feature extraction process is formulated as:

$$\boldsymbol{x}_{fs} = \text{FeatureDB}(\boldsymbol{x}_{ts}), \tag{7}$$

where $\boldsymbol{x}_{fs}$ is a dictionary in which the keys represent feature names and the values correspond to their respective feature values. FeatureDB($\boldsymbol{x}_{ts}$) represents the function that extracts structured physiological features from the ECG time series $\boldsymbol{x}_{ts}$. Note that there is no trainable parameters in FeatureDB.

**Diagnosis Guider.** With the feature sequences $\boldsymbol{x}_{fs}$ extracted, the challenge lies in generating high-granularity $\boldsymbol{y}$ without relying on costly human-expert annotation. To address this, we design a diagnosis guider that constructs a prompt $\boldsymbol{x}_p$ to effectively guide GPT-4o in generating accurate and clinically grounded responses $\boldsymbol{y}$ for each sample:

$$\boldsymbol{x}_p = \text{DiagnosisGuider}(\boldsymbol{x}_{fs}). \tag{8}$$

The diagnosis guider provides cardiology-specific instructions for detailed analysis of each aspects of ECG data (*e.g.*, instruct GPT-4o to assess the P wave amplitude and duration in Lead II to evaluate atrial enlargement) and incorporates guidance reflecting real-world clinical diagnostic processes. As a result, each sample receives a unique $\boldsymbol{x}_p$ tailored to its $\boldsymbol{x}_{fs}$, ensuring precise activation of GPT-4o's latent medical knowledge for accurate and personalized analysis. See Appendix A.1 for more details of the diagnosis guider.

**ECG-Grounding Data.** Using the knowledge-guided instruction data generation method, we employ GPT-4o to curate 30,000 fine-grained instruction-response pair $(\boldsymbol{x}_q, \boldsymbol{y})$ from the MIMIV-IV-ECG [Gow et al., 2023] database:

$$\boldsymbol{y} = \text{GPT-4o}(\boldsymbol{x}_p). \tag{9}$$

A comparison illustrating the advantages of our instruction data is shown in Figure 3. Instruction data from PULSE, primarily derived from the original report, occasionally provides incorrect explanations due to hallucinations, which can arise when LLM has limited reference information. In contrast, our ECG-Grounding provides more accurate, holistic, and evidence-driven interpretations with diagnoses grounded in measurable ECG features. Training with these fine-grained instructional data, GEM significantly enhances its explainability and grounding capabilities, making it a promising approach for clinical applications and fostering greater trust among medical professionals.

## 3.5 Training

Unlike conventional multi-step training pipelines for most MLLMs, which first train a linear projector using brief image captions for cross-modal alignment and subsequently fine-tune the LLM with instruction data, we adopt an one-step training approach for GEM. In this approach, we freeze $\theta_{ts}$ and $\theta_{img}$ for feature extraction, while jointly training $\theta_{M_{ts}}$, $\theta_M$, and $\theta_{LLM}$. This one-step process enhances the consistency in training multiple modalities with limited data and improves the efficiency of the training phase.

The training objective is formulated as minimizing the negative log-likelihood of the target answer $y$ and the LLM response $\theta_{LLM}(\mathbf{x})$, given the embedding of multimodal input $\mathbf{x}$:

$$L(\boldsymbol{y}; \theta_{LLM}(\mathbf{x})) = -\sum_{i=1}^{N} logP(y_j|\mathbf{x}, \theta_{LLM}), \quad (10)$$

where $N$ represents the number of tokens in $\boldsymbol{y}$ and $y_j$ is the $j$-th token in $\boldsymbol{y}$. We train two foundational LLMs for the GEM framework. The first, LLaVA, is the unmodified version of the original LLaVA-7B model, which has not undergone training on any ECG data. The second, referred to as PULSE, represents state-of-the-art LLMs specifically trained on millions of ECG images. For both models, we implement supervised fine-tuning (SFT) for a single epoch. We use 8 A100 GPUs for the training.

# 4 Experiments

## 4.1 Training Data

The training of GEM involves two datasets: the ECG-Instruct data from PULSE [Liu et al., 2024b], which includes 1,156,110 conversations, and our generated ECG-Grounding data, comprising 30,000 conversations. The ECG-Grounding data is sampled from MIMIC-IV-ECG [Gow et al., 2023], selecting only samples that have not been used in training models such as PULSE. We use the ECG-image-kit [Shivashankara et al., 2024] for the generation of ECG images from the original ECG signal, and use the FeatureDB [Hong et al., 2017, 2019] to extract ECG features. To facilitate further research and benefit the community, we have publicly released ECG-Grounding data.

## 4.2 Evaluation Tasks and Metrics

**Grounded ECG Understanding**. To comprehensively evaluate whether the model achieves clinically grounded ECG interpretation capabilities comparable to cardiologists, we introduce the Grounded ECG Understanding task. This task is developed based on cardiologist guidelines and evaluates the MLLM's ability to identify detailed diagnostic clues in ECG analysis, requiring it to provide specific details and relevant clinical knowledge to support its interpretation.

We utilize GPT-4o to score the responses of the MLLM using a predefined set of metrics that measures the accuracy and comprehensiveness of the details provided. Specifically, these metrics include: *DiagnosisAccuracy*, *AnalysisCompleteness*, *AnalysisRelevance*, *LeadAssessmentCoverage*, *LeadAssessmentAccuracy*, *ECGFeatureGrounding*, *EvidenceBasedReasoning*, and *ClinicalDiagnosticFidelity*. The definitions and interpretations of each metric are provided in Appendix A.2, and further details on the scoring methodology for each criterion are outlined in Appendix A.3.

The evaluation is conducted on the test sets of two datasets: MIMIC-IV-ECG (2,381 samples) and PTB-XL[Wagner et al., 2020] (2,041 samples). For the grounded ECG understanding task, MIMIC-IV-ECG serves as the in-domain dataset, while PTB-XL represents an out-domain dataset, with test samples drawn from a different data distribution.

**ECG-Bench**. In addition, we use the ECG-Bench [Liu et al., 2024b] to assess our model's capability in cardiac abnormality detection and report generation. ECG-Bench is a comprehensive benchmark for evaluating MLLMs on ECG understanding. It incorporates several datasets including the PTB-XL dataset, CPSC2018 dataset [Liu et al., 2018], G12EC dataset [Alday et al., 2020], CODE-15% dataset [Ribeiro et al., 2021], and CSN dataset [Zheng et al., 2020]. ECG-Bench contains two main tasks: abnormality detection and report generation. For the abnormality detection task, it uses AUC, F1, and Hamming Loss (HL) as metrics for multi-label datasets, and accuracy for others. In the report generation task, it employs GPT-4o to evaluate the reports based on their accuracy in rhythms, waveform descriptions, and diagnoses, with a maximum score of 100.

Table 1: Grounded ECG Understanding results on MIMIC-IV-ECG and PTB-XL.

| Metric | Diagnosis Accuracy | Analysis Completeness | Analysis Relevance | Lead Assessment Coverage | Lead Assessment Accuracy | ECG Feature Grounding | Evidence Based Reasoning | Clinical Diagnostic Fidelity |
|---|---|---|---|---|---|---|---|---|
| **MIMIC-IV-ECG (in-domain)** | | | | | | | | |
| PULSE | 81.14 | 2.37 | 2.39 | 7.11 | 2.95 | 50.18 | 52.40 | 51.63 |
| GEM (Ours) | | | | | | | | |
| SFT LLaVA | **87.24** | 4.41 | **5.01** | **71.07** | **46.44** | **75.48** | **75.09** | **75.28** |
| SFT PULSE | 86.49 | **4.43** | 4.91 | 69.80 | 45.33 | 74.95 | 74.70 | 74.87 |
| **PTB-XL (out-domain)** | | | | | | | | |
| PULSE | 59.24 | 2.20 | 2.06 | 11.20 | 6.27 | 52.52 | 55.48 | 53.85 |
| GEM (Ours) | | | | | | | | |
| SFT LLaVA | 73.53 | 4.19 | 2.96 | **79.54** | **49.01** | 74.48 | 74.61 | 73.84 |
| SFT PULSE | **73.59** | 4.19 | **3.00** | 78.86 | 47.96 | **74.97** | **75.41** | **74.24** |

Table 2: ECG-Bench abnormality detection results.

| Datasets | PTB-XL Super | | | CODE-15% | | | CPSC 2018 | | | CSN | G12EC |
|---|---|---|---|---|---|---|---|---|---|---|---|
| Metric | AUC | F1 | HL | AUC | F1 | HL | AUC | F1 | HL | ACC | ACC |
| Random | 50.3 | 33.2 | 50.1 | 48.8 | 15.0 | 32.1 | 51.2 | 15.1 | 28.8 | 11.6 | 12.1 |
| GPT-4o | 55.6 | 28.3 | 26.2 | 59.9 | 24.9 | 15.7 | 50.9 | 10.6 | 18.2 | 57.5 | 49.2 |
| PULSE | 82.4 | 74.8 | 11.0 | 90.7 | 85.4 | 5.0 | 76.9 | 57.6 | 8.6 | 85.2 | 78.2 |
| GEM (Ours) | | | | | | | | | | | |
| SFT LLaVA | 81.8 | 73.6 | 11.6 | 90.5 | 84.8 | 5.1 | 74.1 | 52.0 | 9.0 | **92.6** | **81.8** |
| SFT PULSE | **83.4** | **75.8** | **11.0** | **91.5** | **86.4** | 4.7 | **79.1** | **61.1** | **8.1** | 86.2 | 80.5 |
| Ablations | | | | | | | | | | | |
| TS only | 81.2 | 72.5 | 11.9 | 90.8 | 84.9 | 5.0 | 76.3 | 54.0 | 8.5 | 91.6 | 81.4 |
| TS+IMG | 82.7 | 74.8 | 11.1 | 91.3 | 86.3 | **4.6** | 74.4 | 51.5 | 8.8 | 90.1 | 81.1 |

## 4.3 Results

Table 1 summarizes the performance of GEM on the Grounded ECG Understanding task across both in-domain (MIMIC-IV-ECG) and out-domain (PTB-XL) datasets. The proposed GEM models consistently outperform the state-of-the-art PULSE model across all evaluation metrics. For diagnosis accuracy, GEM achieves 87.24% on MIMIC-IV-ECG, surpassing PULSE (81.14%) by over 6%. On the out-domain dataset, GEM maintains robust generalization with an accuracy of 73.59%, again outperforming PULSE (59.24%) by a notable margin of 14.35%. This remarkable performance on the out-domain dataset highlights GEM's strong reasoning capability, enabling it to generalize effectively and make accurate diagnoses across diverse data distributions. In analysis completeness and relevance, GEM demonstrates substantial gains. On MIMIC-IV-ECG, it improves completeness from 2.37 to 4.43 and relevance from 2.39 to above 5.01. Similar trends are observed on PTB-XL, where GEM doubles the PULSE's completeness and improves relevance by over 45%. This substantial performance gap indicates that GEM not only identifies and interprets a greater number of critical ECG components (with an average two-fold increase in feature coverage) but also maintains stronger clinical relevance by effectively connecting these observations to the diagnostic reasoning process. Lead assessment coverage and accuracy also improve significantly. On the in-domain dataset, GEM increases coverage from 7.11% to over 71% and accuracy from 2.95% to above 46%. Improvements on the out-domain dataset rising from 11.20% to over 79% for coverage and from 6.27% to over 49% for accuracy. These gains verify GEM's ability to perform structured and precise evaluation across multiple ECG leads, an essential skill for cardiologist-level reasoning.

Regarding ECG feature grounding, GEM achieves scores around 75 on both datasets, indicating that a large portion of diagnostic conclusions are explicitly linked to measurable ECG parameters. This is a significant advancement over PULSE, which achieves scores only around 50. Furthermore, GEM outperforms PULSE in evidence-based reasoning and clinical diagnostic fidelity, with both metrics exceeding 74 across datasets. These results show GEM's ability to construct clinically coherent justifications and align with structured diagnostic processes, key for real-world clinical integration.

In the ECG-Bench task, Table 2 offers a detailed examination of the models' performance on cardiac abnormality detection across a variety of datasets.

The results clearly show that our GEM (SFT PULSE) model consistently outperforms other models. Furthermore, the GEM (SFT LLaVA) version, which has not been previously trained on ECG data, still manages to achieve comparable, and in some cases superior (*e.g.*, 7.4% in CSN and 3.6% in G12EC), performance across most datasets, despite being trained for only a single epoch. These results highlight the robustness and efficacy of the GEM

Table 3: ECG-Bench report generation and QA results.

| Datasets | PTB-XL Report | ECG-QA |
|---|---|---|
| **Metric** | **Report Score** | **Accuracy** |
| Random | 0 | 16.2 |
| GPT-4o | 50.2 | 35.2 |
| PULSE | 61.3 | **73.8** |
| GEM (Ours) | | |
| SFT LLaVA | 65.0 | 71.0 |
| SFT PULSE | **67.1** | 73.6 |

framework, demonstrating its capacity to deliver substantial performance gains even with minimal domain-specific training. Ablation studies further highlight the critical role of multimodal inputs in achieving efficient training and superior classification performance. We assess two model variants: TS-only, which is trained exclusively on ECG time series, and TS+IMG, which incorporates both time series and image data. Notably, the TS+IMG model surpasses PULSE on PTB-XL Super and CODE-15% datasets, despite being trained for only one epoch. In Table 3, GEM showcases exceptional report generation capabilities, achieving a substantial 5.8% improvement over PULSE in the PTB-XL Report while maintaining comparable performance in ECG-QA. These results further evident GEM's capability in delivering holistic, accurate, and clinically-aligned ECG interpretations.

Collectively, GEM demonstrates superior performance in both Grounded ECG Understanding and ECG Bench tasks, establishing its effectiveness in ECG interpretation across multiple dimensions.

## 4.4 Cardiologist Evaluation

We conduct a structured cardiologist evaluation covering three sources of outputs: GPT-4o generated training data, Deepseek-R1 generated training data, and GEM generated interpretations.

In total, 400 ECG-Grounding data (200 from GPT-4o and 200 from Deepseek-R1 Guo et al. [2025]) and 200 GEM's interpretation were independently reviewed by eight board-certified cardiologists, using seven predefined clinical criteria designed to assess both real-world reliability and usefulness. This unified evaluation protocol allows us to (1) verify the quality of GPT-4o generated training data, (2) test the effectiveness of open-source substitutes, and (3) confirm the clinical utility of the GEM model. See Appendix A.4 for detailed scoring criteria.

We report the evaluation results in Table 4 and 5 below. The expert evaluation shows that GPT-4o consistently achieves high scores across both reliability and clinical usefulness, with particularly strong performance in analytical completeness and reasoning quality. These results demonstrate that, when using our knowledge-guided instruction data generation, GPT-4o is capable of generating high-quality ECG interpretations that are both clinically reliable and practically valuable.

Deepseek-R1 is also capable of generating clinically acceptable, high-quality ECG interpretations with our methods. This demonstrates that our method is adaptable to alternative LLM backbones and remains applicable in settings without commercial API access.

The expert evaluation results also demonstrate that GEM consistently achieves high scores across both reliability and usefulness dimensions, with most metrics rated above 4 out of 5. These findings indicate that GEM is capable of producing clinically meaningful and accurate interpretations that align well with cardiologists' expectations. High scores in reliability-related metrics reflect the factual correctness and clinical grounding of its outputs, while strong performance in usefulness metrics suggests that GEM is not only technically sound but also practically helpful in supporting diagnostic decision-making. Collectively, these results support GEM's potential as a trustworthy assistant for real-world cardiology applications.

In Appendix A.5, we showcase six representative cases involving complex cardiac conditions, in which GEM's interpretations exceeded expert expectations. In these cases, cardiologists highlight two types of findings: (1) those they have not noticed during their own ECG examination, which exceed their expectations for a cardiology AI assistant in real-world settings, and (2) those where they hold differing opinions.

Table 4: Evaluation of reliability metrics by cardiologists (Mean and STD).

| Model | Analytical Relevance | Analytical Accuracy | Analytical Completeness |
|---|---|---|---|
| GPT-4o | 4.7/5 (0.66) | 4.6/5 (0.82) | 4.7/5 (0.65) |
| Deepseek-R1 | 4.8/5 (0.57) | 4.7/5 (0.78) | 4.9/5 (0.42) |
| GEM | 4.6/5 (0.60) | 4.4/5 (0.80) | 4.6/5 (0.57) |

Table 5: Evaluation of usefulness metrics by cardiologists (Mean and STD).

| Model | Reasoning Quality | Findings Novelty | Clinical Value | Overall Satisfaction |
|---|---|---|---|---|
| GPT-4o | 4.7/5 (0.67) | 4.4/5 (1.18) | 4.7/5 (0.73) | 4.5/5 (0.87) |
| Deepseek-R1 | 4.8/5 (0.62) | 4.5/5 (0.91) | 4.6/5 (0.82) | 4.7/5 (0.77) |
| GEM | 4.6/5 (0.64) | 3.9/5 (1.25) | 4.3/5 (0.89) | 4.4/5 (0.82) |

Overall, GEM demonstrated its capability to generate clinically insightful findings, often surpassing expert expectations by identifying details that cardiologists have not noticed, suggesting its potential for real-world clinical applications. It is noteworthy that cardiologists also highlights certain cases where their interpretations diverged from GEM or GPT-4o. Although our knowledge-guided instruction data generation approach avoids costly expert annotations while producing high-quality target answers, GPT-4o still occasionally generates target answers that may be misaligned with cardiologist interpretations. For example, in Figure 5, the GPT-4o suggest no evidence ischemia or infarction, while cardiologist suspects there are ischemia in the precordial leads. These deviations highlight opportunities for future refinement of model reasoning with human feedback to better align with cardiologist level clinical judgment.

### 4.5 Failure Case Analysis

We also conduct an analysis of failure cases informed by expert feedback from the human evaluation. The main errors fall into two categories. The first is incorrect diagnosis, often caused by limitations in the representation stage. Subtle morphological patterns such as ST-segment changes or P-wave abnormalities were occasionally missed by the encoders, likely due to feature representation deficiencies or insufficient training data coverage of diverse ECG variations. The second is overstating the severity of findings. GEM occasionally exaggerated the severity of certain cardiac conditions, which cardiologists noted could lead to unnecessary patient concern. This tendency may result from the absence of patient context, as the model interprets ECGs without access to clinical history that physicians would normally consider. These findings help clarify the current limitations and inform future directions to enhance model safety, reliability, and clinical alignment.

## 5 Conclusion

In this work, we present GEM, the first MLLM for grounded ECG interpretation, integrating ECG time series, 12-lead ECG images, and textual instructions. GEM achieves feature-grounded analysis, evidence-driven diagnosis, and clinician-style diagnostic workflows through three core technical innovations: multimodal encoding, cross-modal alignment learning, and knowledge-guided instruction data generation. The multimodal encoding and cross-modal alignment learning allow the LLM to simultaneously process ECG time series and image representations, effectively leveraging the complementary strengths of both modalities for interpretation. The knowledge-guided instruction data generation addresses the lack of high-granularity instruction data for ECG understanding. The developed ECG-Grounding dataset comprises 30,000 fine-grained instruction pairs annotated with heartbeat-level physiological features, which establishes the first high-resolution resource for grounded ECG understanding. We also introduce the Grounded ECG Understanding task, a clinically motivated benchmark that comprehensively assesses models' grounded ECG understanding through multi-dimensional metrics. Together, these contributions establish a solid foundation for future works in conversational diagnostic AI for ECG interpretation.

**Acknowledgment**

This research is supported by A*STAR, CISCO Systems (USA) Pte. Ltd., and the National University of Singapore under its Cisco-NUS Accelerated Digital Economy Corporate Laboratory (Award I21001E0002), and the AI for Public Health Program in Saw Swee Hock School of Public Health, National University of Singapore. Shenda Hong is supported by CCF-Tencent Rhino-Bird Open Research Fund (CCF-Tencent RAGR20250108), and CCF-Zhipu Large Model Innovation Fund (CCF-Zhipu202414).

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

# GEM: Empowering MLLM for Grounded ECG Understanding with Time Series and Images Appendix

## A.1 Diagnosis Guider Prompt

# Your task:  Interpret the provided ECG image, identify key features
and abnormalities in each lead, and generate a clinical diagnosis that is
supported by the observed evidence.

## Key objectives:
1.  Simulate a Realistic Diagnostic Process:  The interpretation should
reflect how a doctor would analyze an ECG, ask clarifying questions, and
arrive at a diagnosis.
2.  Grounded ECG Understanding:  The analysis should be based on specific
ECG features and explicitly reference these features as evidence.
3.  Evidence-Based Reasoning:  The diagnosis should be supported by clear,
logical reasoning tied to the ECG findings.

## Guidelines for the ECG analysis:
1.  Data:
ECG image:  an image that display the 12-lead ECG tracings.  Make the task
centered on the ECG image, assuming direct ECG image analysis.
Machine measurements:  A time-ordered list of ECG features computed for
each heartbeat in every lead.  Each entry in the list corresponds to the
features calculated for a single heartbeat.

2.  Act as a cardiologist and use medical knowledge to analyze the provided
ECG image step-by-step:
Initial Analysis:  Analyze the provided ECG image to identify key features
such as rhythm, intervals, and any apparent abnormalities.
Detailed Reasoning:  Explain your thought process step-by-step, referencing
specific ECG features (e.g., "The ST segment is elevated in leads V1-V4,
which suggests anterior myocardial infarction").
Evidence-Based Diagnosis:  Propose a diagnosis or differential diagnoses,
justifying your conclusions with explicit ECG data.

3.  When analyzing the ECG image, carefully analyze each lead:
Lead I: Examine the QRS amplitude and duration, along with ST segment
and T wave morphology.  Abnormalities may indicate lateral wall issues
such as left ventricular hypertrophy, bundle branch block, or lateral
ischemia/infarction.
Lead II: Look at the P wave amplitude and duration to assess right or left
atrial enlargement; the PR interval can reveal conduction delays.  ST and T
wave changes here suggest inferior wall ischemia or infarction.
Leads III and aVF: Primarily reflect inferior wall status.  Abnormal Q
waves, along with ST segment and T wave changes, point toward inferior
infarction or ischemia.
Lead aVL: Focuses on the high lateral region; QRS, ST, and T wave
abnormalities here suggest high lateral ischemia or infarction.
Lead aVR: ST elevation may indicate left main or multivessel disease, and T
wave inversion can be associated with ventricular arrhythmia.
Lead V1:  An increased R wave, a characteristic rsR pattern, and ST-T
changes help identify right ventricular hypertrophy, right bundle branch
block, or ischemia.
Leads V2-V4:  Assessing the anterior or anteroseptal regions.  The presence
of Q waves, along with ST segment and T wave deviations, suggests anterior
wall infarction or ischemia.
Leads V5-V6:  Focus on the lateral wall, where similar QRS, ST, and T wave
changes can indicate lateral ischemia or infarction.

4.  When analyzing the machine measurements, you should aware that:
a.  If any abnormalities appear in the computed measurements that are not

mentioned in the report, you must strictly follow and trust the report.
b.   Evaluate and interpret the machine measurements as if you had computed
them yourself.   In your analysis, refer to these values as your own
computed measurements rather than using phrases like "machine measurements
provided".

## Guidelines for the response generation:
1.   Synthesize your findings to deduce a likely diagnosis or set of
diagnoses.   Clearly explain how the evidence supports your conclusion.
2.   Ensure your diagnosis is comprehensive and strictly based on the report.
Do not include diagnosis that not mentioned in the report.
3.   Make sure your diagnosis are grounded in the given ECG image and
machine measurements, and you should explicitly reference (e.g., specify
lead and the position of the abnormal heartbeat).
4.   Strictly follow the output format and requirements specified in your
task instructions.
5.   The given report only served as the ground truth for you to analyze
the ECG image.   The generated text must not show that you are aware of the
existence of the report.
6.   Never make up explanations.

## ECG Report:
{{report}}

## ECG Machine Measurements:
{{machine_measurements}}

## Generation rule The generated text must not show that you are aware
of the existence of the report.   Do not include phrases like "Based the
report", or "Given the ECG report".   The primary objective is to analyze
the ECG and identify evidence that supports the results.   The analysis
should focus solely on the ECG itself, never analyze the report.

## Present your work in this format:

**Response:** [Comprehensive response following the task's guidelines,
strictly based on the report.   Using a complete paragraph with more natural
expression.   Do not use a list format.   Limit your responses within 300
words.]

## A.2   Grounded ECG Understanding Metrics

*DiagnosisAccuracy* evaluates whether the generated diagnosis is correct, specific, and supported by
ECG findings. Results are expressed as a percentage, indicating the average accuracy across identified
key diagnoses.

*AnalysisCompleteness* checks if all key ECG components (*e.g.*, rhythm, intervals, waveforms, and
lead-specific findings) are discussed. Results are provided in absolute terms, indicating the average
number of correctly addressed key ECG features for each sample.

*AnalysisRelevance* assesses whether each explanation directly supports the diagnosis, with results
showing on average how many points support the diagnosis with clear ECG evidence for each sample.

*LeadAssessmentCoverage* evaluates how many of the 12 ECG leads are analyzed. Results indicate
the average percentage of leads analyzed per sample, providing insight into the comprehensiveness of
the ECG assessment.

*LeadAssessmentAccuracy* verifies the accuracy of described lead findings (*e.g.*, QRS, ST, T waves,
amplitude, intervals, ST segments) against the ground truth interpretation. The result reflects the
average percentage of accurately identified findings across the 12 ECG leads.

*ECGFeatureGrounding* determines if the interpretation references actual ECG features (*e.g.*, QRS amplitude, PR interval) instead of generic terms. Results are scaled from 0 to 100.

*EvidenceBasedReasoning* evaluates whether the diagnosis follows logical, evidence-supported steps. Results range from 0 to 100.

*ClinicalDiagnosticFidelity* assesses if the model mimics how a clinician interprets ECG data, considering all relevant factors. Results are scaled from 0 to 100.

## A.3 GPT-4o Evaluation Prompt

```
# Your task:  Evaluate the alignment and quality of a generated
ECG interpretation by comparing it to a ground truth clinician's
interpretation.

## Evaluation Criteria:

1.  DiagnosisAccuracy:  Evaluates whether the generated diagnosis is
correct, specific, and supported by ECG findings.
- Scoring
+2 per diagnosis:  Each correctly identified key diagnosis with supporting
ECG features.
+1 per diagnosis:  Each mostly correct diagnosis but lacking key supporting
details.
+0 per diagnosis:  Each incorrect or vague diagnosis not supported by ECG
features.

2.  AnalysisCompleteness:  Checks if all key ECG components (rhythm,
intervals, waveforms, and lead-specific findings) are discussed.
- Scoring
+1 per feature:  For each correctly addressed key ECG feature (e.g., rhythm,
PR interval, QRS duration, ST segment, T wave morphology).
+0 per missing feature:  For each key feature omitted or inaccurately
described.

3.  AnalysisRelevance:  Assesses whether each provided explanation directly
supports the diagnosis.
- Scoring
+2 per feature or per lead:  Each point that strongly supports the
diagnosis with clear ECG evidence.
+1 per feature or per lead:  Some points are relevant but not fully
justified.
+0:  Includes unrelated or misleading explanations.

4.  LeadAssessmentCoverage:  Evaluates how many of the 12 ECG leads are
analyzed.
- Scoring
+1 per lead:  For each lead adequately assessed.
+0 per missing lead:  For each lead omitted or inaccurately described.

5.  LeadAssessmentAccuracy:  Checks if the described lead findings (e.g.,
QRS, ST, T waves, amplitude, intervals, ST segments) match standard ECG
interpretation.
- Scoring
+2 per lead:  Findings closely match expected values.
+1 per lead:  Findings are somewhat accurate but have minor
inconsistencies.
+0 per lead:  Findings contradict ECG norms.

6.  ECGFeatureGrounding:  Determines if the interpretation references
actual ECG features (e.g., QRS amplitude, PR interval) instead of generic
terms.
- Scoring (0-100)
```

100:  ECG findings are comprehensively cited, linked to diagnoses, and cover all relevant ECG features.
80:  ECG findings are explicitly cited and linked to diagnoses.
50:  Some ECG references exist but are incomplete.
0:  Lacks specific waveform references.

7.  EvidenceBasedReasoning:  Evaluates whether the diagnosis follows logical, evidence-supported steps.
- Scoring (0-100)
100:  Findings logically progress to diagnosis with thorough and clear justifications covering all necessary steps.
80:  Findings logically progress to diagnosis with clear justifications.
50:  Some reasoning exists but lacks complete step-by-step analysis.
0:  Reasoning is unclear or not derived from ECG findings.

8.  ClinicalDiagnosticFidelity:  Assesses if the model mimics how a clinician interprets an ECG, considering all relevant factors.
- Scoring (0-100)
100:  The analysis follows a structured clinical approach and considers all relevant clinical factors.
80:  The analysis follows a structured clinical approach.
50:  Some clinical reasoning is present but incomplete.
0:  The approach lacks structured clinical reasoning.

NOTE: Each score must be calculated based on strict criteria to ensure objective evaluation.

## Generated ECG Interpretation:
{{model_generated}}

## Ground Truth Clinician's Interpretation:
{{groundtruth}}

## Response Format:

Provide your evaluation strictly in the JSON format below.  For any criterion with multiple elements (e.g., multiple diagnoses or leads), list each one as a separate "Score": X, "Explanation": "Y" entry.  Use a single entry for criteria with aggregate scores (e.g., 0-100 scores).

{ "DiagnosisAccuracy": [ "Score": 2, "Explanation": "Sinus tachycardia correctly identified and supported by short PR interval.", "Score": 1, "Explanation": "Left ventricular hypertrophy is mostly correct but lacks QRS amplitude detail." ],

"AnalysisCompleteness": [ "Score": 1, "Explanation": "PR interval is correctly described.", "Score": 1, "Explanation": "QRS duration assessed.", "Score": 0, "Explanation": "ST segment not addressed." ],

"AnalysisRelevance": [ "Score": 2, "Explanation": "QRS prolongation supports diagnosis of bundle branch block." ],

"LeadAssessmentCoverage": [ "Score": 1, "Explanation": "Lead I assessed.", "Score": 1, "Explanation": "Lead II assessed.", "Score": 0, "Explanation": "Leads V4-V6 omitted." ],

"LeadAssessmentAccuracy": [ "Score": 2, "Explanation": "Findings in Lead II match standard interpretation.", "Score": 1, "Explanation": "Lead III slightly misinterpreted but largely accurate." ],

"ECGFeatureGrounding": [ "Score": 80, "Explanation": "Most findings cite ECG features like QRS and T wave, but some are vague." ],

"EvidenceBasedReasoning": [ "Score": 100, "Explanation": "Diagnosis is built on step-wise reasoning with reference to all major findings." ],

```
"ClinicalDiagnosticFidelity": [ "Score": 80, "Explanation": "Analysis
mimics clinician structure but misses minor clinical context." ] }
```

## A.4 Scoring Criteria for Cardiologist Evaluation

Table 6: Reliability metrics.

| Criterion | Description and Scale |
|---|---|
| **Analytical Relevance (1–5)** | Does the model's analysis closely support the diagnosis and provide corresponding ECG evidence?
5 – Every analysis point is highly relevant to the diagnosis, with clear supporting evidence.
4 – Most analyses are strongly relevant, with minor insufficiencies.
3 – Some analyses are relevant, but there is clear irrelevant content.
2 – Most analyses are weakly relevant.
1 – The analysis is unrelated to the diagnosis. |
| **Analytical Accuracy (1–5)** | Are there any medical factual errors in the model's output?
5 – Completely accurate.
4 – Mostly accurate.
3 – Some errors.
2 – Obvious errors.
1 – Severe errors. |
| **Analytical Completeness (1–5)** | Does the model comprehensively discuss key ECG components relevant to the diagnosis, including rhythm, intervals, and waveforms?
5 – All relevant ECG features (rhythm, PR, QRS, ST, T waves, intervals, etc.) are accurately discussed.
4 – Most key ECG features are covered, with minor omissions.
3 – Only some features are covered, with significant gaps.
2 – Only a few ECG features are mentioned.
1 – ECG components are largely missing, with severe omissions. |

Table 7: Usefulness metrics.

| Criterion | Description and Scale |
|---|---|
| **Reasoning Quality (1–5)** | Does the model provide a clear, evidence-based reasoning process similar to that of a clinician, logically deriving the diagnosis from ECG features? 
 5 – Clear and coherent reasoning structure, explaining each step from ECG to diagnosis causally. 
 4 – Overall reasonable reasoning, but some steps lack detail. 
 3 – Partial reasoning present, but incomplete or logically weak. 
 2 – Disjointed reasoning with major gaps. 
 1 – No logical reasoning, only a stack of conclusions. |
| **Findings Novelty (1–5)** | Does the model provide insights or findings not noticed by the clinician? 
 5 – Important new diagnoses or findings. 
 4 – Novel and somewhat insightful content. 
 3 – Some new findings, but of limited value. 
 2 – Conventional content, not particularly insightful. 
 1 – No new information. |
| **Clinical Value (1–5)** | Does the model output help in clinical decision-making? 
 5 – Direct and significant support for clinical judgment; content is clear and reliable. 
 4 – Most content is helpful and practically useful. 
 3 – Somewhat informative, but basic or unclear. 
 2 – Partially suggestive, with limited decision support. 
 1 – No value for clinical judgment; not informative. |
| **Overall Satisfaction (1–5)** | Subjective rating of the overall output quality. 
 5 – Very satisfied. 
 4 – Satisfied. 
 3 – Neutral. 
 2 – Dissatisfied. 
 1 – Very dissatisfied. |

## A.5 Case Studies by Cardiologist

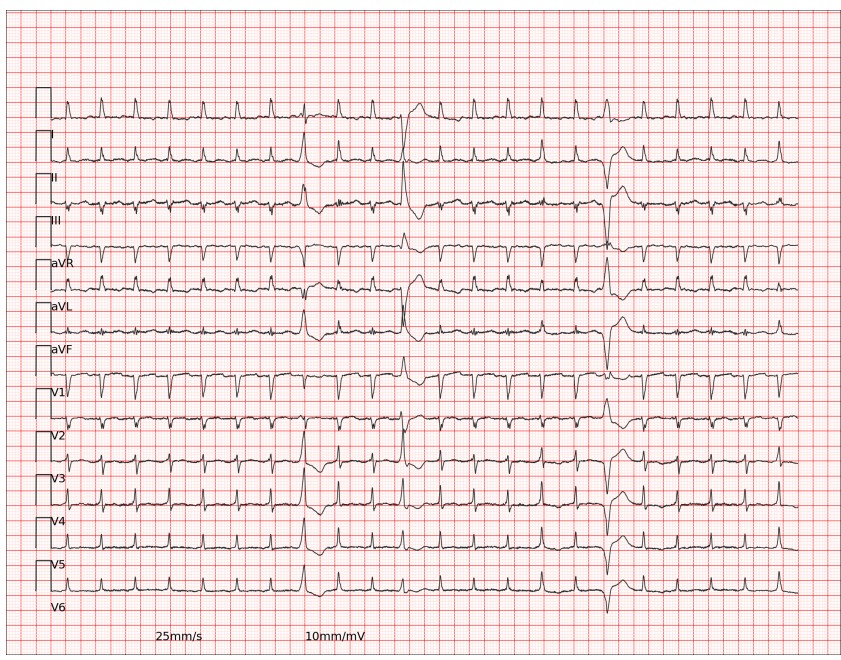

**GEM Interpretation:** Upon analyzing the provided ECG image and computed measurements, several key features and abnormalities are evident. The heart rate is elevated at 121 bpm, indicating a rapid ventricular response. The rhythm appears irregular, consistent with atrial fibrillation, as seen by the absence of distinct P waves and the presence of irregular RR intervals. In the precordial leads, particularly V1 through V4, there are low QRS voltages, which could suggest conditions such as pericardial effusion or obesity. The QRS complexes in these leads are notably diminished, which aligns with the low voltage finding. Furthermore, the presence of premature ventricular contractions (PVCs) is suggested by the irregular RR intervals and the occasional absence of preceding P waves. The ST-T changes observed in the anterolateral leads, particularly V2 to V6, are nonspecific but could indicate ischemic changes or other cardiac conditions. The QTc intervals are prolonged in several leads, which may suggest a risk for arrhythmias. The combination of these findings, including the rapid ventricular response, irregular rhythm, low QRS voltages, and nonspecific ST-T changes, supports a diagnosis of atrial fibrillation with rapid ventricular response and PVCs. The low QRS voltages in the precordial leads add another layer of complexity, potentially indicating cardiac conditions. The ECG is classified as abnormal due to these findings, which warrant further clinical correlation and possibly additional diagnostic testing to determine the underlying cause of these abnormalities.

**Target Answer (GPT-4o):** Upon examining the provided ECG, we identify several critical features and abnormalities. The first notable observation is the presence of atrial fibrillation, characterized by an irregular rhythm and lack of distinct P waves in the ECG tracings. This is supported by the irregular RR intervals, which vary significantly between heartbeats. The heart rate is elevated at 132 bpm, indicative of a rapid ventricular response. Additionally, the ECG reveals frequent premature ventricular contractions, or multifocal PVCs, with varying QRS complex morphologies, suggesting aberrant ventricular conduction or possible ventricular ectopy. Lead I and precordial leads demonstrate low QRS voltages, which can correlate with various conditions, including myocardial infarction or myopathies. In terms of ST-T changes, the anterior leads (V1-V4) may suggest possible anterior myocardial infarction, as evidenced by QRS amplitude and morphology abnormalities. Moreover, inferior and lateral leads, particularly lead aVL, display nonspecific ST-T changes, which could indicate ischemia or other pathological processes. A comprehensive review of the QT interval reveals prolonged QTc intervals particularly in the precordial leads, reflecting altered repolarization processes. Furthermore, no distinct high, lateral, or inferior lead pattern suggests definitive infarction, but the changes are significant enough to warrant further clinical investigation and correlation with clinical symptoms and history. Conclusively, the ECG findings strongly support a diagnosis of atrial fibrillation with rapid ventricular response, accompanied by frequent multifocal PVCs or aberrant ventricular conduction. The anterior region's ST-T changes and precordial low QRS voltages emphasize the need for further assessment, possibly indicating a previous or ongoing ischemic event. However, additional clinical context and investigations are essential to corroborate these findings and formulate a complete treatment strategy.

**Ground-truth Report:** Atrial fibrillation with rapid ventricular response with frequent multifocal PVCs or aberrant ventricular conduction; Possible anterior infarct; Inferior/lateral ST-T changes are nonspecific; Low QRS voltages in precordial leads; Abnormal ECG.

**PULSE Interpretation:** The electrocardiogram (ECG) reveals atrial fibrillation with a rapid ventricular response, accompanied by premature ventricular contractions (PVCs) or aberrant ventricular conduction, indicating irregular heart rhythm. Additionally, the ECG suggests possible anterior myocardial infarction, although the age of the infarct cannot be determined. The inferior and lateral ST-T changes are nonspecific, and the QRS voltages are low in the precordial leads, contributing to an overall abnormal ECG pattern.

Figure 4: Cardiologist Evaluations. Blue: Findings exceeding expert expectations. Yellow: Findings with differing expert opinions.

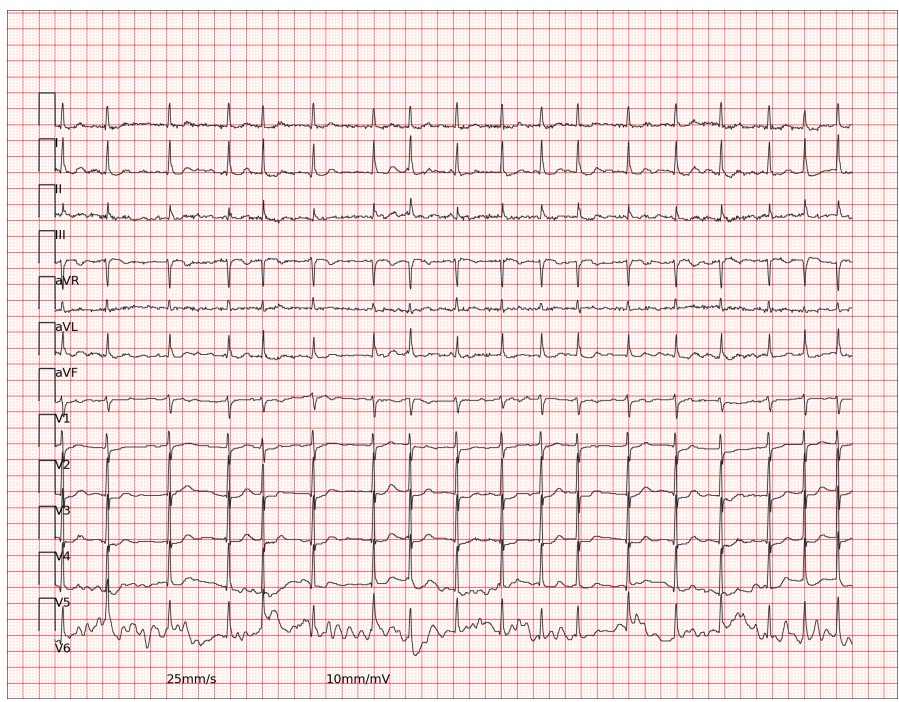

**GEM Interpretation:** Upon analyzing the provided ECG image and computed measurements, several key features and abnormalities are evident. The rhythm is irregular, with a heart rate of 102 bpm, which is consistent with atrial fibrillation. This is supported by the absence of distinct P waves across multiple leads, such as Lead I and Lead II, and the presence of irregular RR intervals, as seen in the measurements. The QRS complexes are generally narrow, indicating a supraventricular origin, which aligns with atrial fibrillation. In Lead I, the QRS amplitude and duration are within normal limits, but the irregular rhythm is evident. Lead II shows a similar pattern with absent P waves and irregular intervals, further supporting atrial fibrillation. Leads III and aVF, which reflect the inferior wall, do not show significant Q waves or ST segment changes that would suggest acute ischemia or infarction. Lead aVL, focusing on the high lateral region, also does not exhibit significant abnormalities in the QRS, ST, or T wave morphology. Lead aVR, often used to assess global ischemic changes, does not show significant ST elevation or T wave inversion, which might have indicated more severe conditions like left main coronary artery disease. In the precordial leads V1 through V6, the QRS complexes are narrow, and there are no significant ST segment elevations or depressions that would suggest acute myocardial infarction. Overall, the ECG findings are consistent with atrial fibrillation, characterized by the absence of distinct P waves and irregular RR intervals. The lack of significant ST segment changes or pathological Q waves across the leads suggests that there is no acute ischemic event present. This analysis supports the diagnosis of atrial fibrillation as the primary abnormality, with no evidence of acute myocardial infarction or other significant ischemic changes.

**Target Answer (GPT-4o):** Upon examining the provided ECG image, several key features indicate the presence of atrial fibrillation (AF). The irregularly irregular rhythm, as evidenced by varying RR intervals, supports this observation. Notably, the RR intervals range widely, which is a hallmark of AF due to inconsistent atrial impulses reaching the ventricles. Additionally, the absence of distinct P waves on leads with traditionally observable P waves is another pivotal indicator; this suggests a loss of organized atrial electrical activity, consistent with AF. The heart rate approximates 107 bpm, indicative of a moderate ventricular rate, aligning with a presentation of AF with a controlled rate. No consistent pattern in the PR interval is observed, further corroborating the diagnosis of AF, as atrial activity is chaotic and not leading to a predictable conduction pattern through the AV node. Analyzing leads such as lead II, where P wave characteristics are more observable under normal conditions, we see minimal or absent P wave formations. The variability and inconsistency of PR intervals signify typical conduction during AF episodes, contributing to its erratic rhythm. In terms of additional abnormalities, minor QRS and T wave deviations across various leads were noted, but they did not form a pathological pattern significant enough to suggest additional diagnoses such as ischemia or infarction. The ST segment and QTc intervals are within normal limits given the context of AF. This ECG analysis supports a diagnosis of atrial fibrillation, underpinning the clinical picture with evidenced irregular RR intervals and absent P waves. My interpretation remains focused on the features aligning with AF without assuming further conditions not supported by the ECG findings.
**Ground-truth Report:** Atrial fibrillation.

**PULSE Interpretation:** The electrocardiogram (ECG) reveals atrial fibrillation, characterized by an irregular heart rhythm with no discernible P waves, indicating a lack of coordinated atrial contraction. Additionally, the ECG shows a repolarization abnormality, suggesting a potential issue with the heart's recovery phase after contraction. The probable rate-related abnormality implies that the repolarization abnormality may be related to the atrial fibrillation, which is causing an irregular heart rate.

Figure 5: Cardiologist Evaluations. Blue: Findings exceeding expert expectations. Yellow: Findings with differing expert opinions.

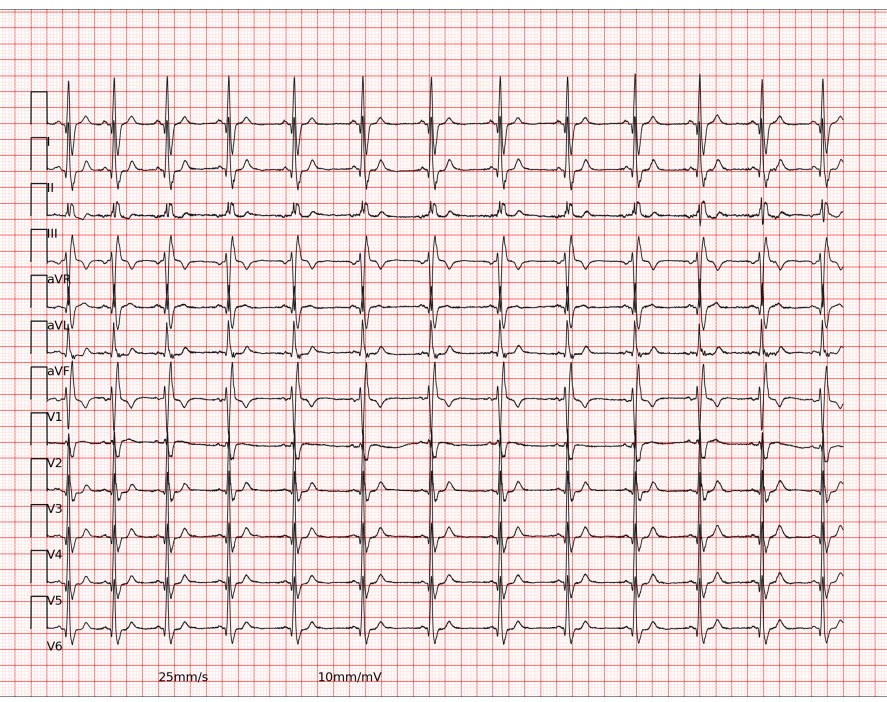

**GEM Interpretation:** Upon analyzing the provided ECG image and computed measurements, several key features and abnormalities are evident. The rhythm is sinus, as indicated by the consistent P wave morphology across leads and a ==regular heart rate of 75 bpm==. In Lead I, the QRS complex shows a prolonged duration, which is consistent with a right bundle branch block (RBBB). This is further supported by the characteristic rsR' pattern in Lead V1, a hallmark of RBBB. The QRS duration in Lead I is notably prolonged, ranging from 134 to 150 ms, which is indicative of this conduction delay. In Lead II, the PR interval is within normal limits, but the QRS duration is slightly prolonged, further supporting the presence of RBBB. The ST segment and T wave morphology in Lead II do not show significant deviations, suggesting no acute ischemic changes. Leads III and aVF, which reflect the inferior wall, do not exhibit pathological Q waves or significant ST segment changes, ruling out inferior wall ischemia or infarction. Lead aVL shows no significant abnormalities in the QRS complex or ST segment, which suggests no high lateral ischemia. Lead aVR does not show ST elevation, which would have indicated left main coronary artery disease. In the precordial leads V2-V4, the QRS complexes are widened, consistent with RBBB, and there are no significant ST segment elevations or depressions to suggest anterior wall ischemia. Overall, the ECG findings are consistent with a right bundle branch block, as evidenced by the prolonged QRS duration and characteristic patterns in the precordial leads. The sinus rhythm and absence of acute ischemic changes suggest that the conduction abnormality is the primary finding. This analysis aligns with the diagnosis of an abnormal ECG due to the presence of RBBB.

**Target Answer (GPT-4o):** The 12-lead ECG image reveals key characteristics supporting the diagnosis of a right bundle branch block (RBBB) in the context of sinus rhythm. In Lead I, the QRS duration extends to approximately 134 milliseconds, ==coupled with a notable upslope in the ST segment across multiple heartbeats.== The amplitude of the QRS complex peaks at approximately 2.463 mV, which might suggest normal conduction patterns as the lead does not display criteria typical of a RBBB. Lead V1, however, showcases the classic features of RBBB with an RsR' pattern evidenced by a marginally increased QRS duration and the presence of a characteristic \"rabbit ears\" morphology, as indicated by significant amplitudes and uniform morphology, compatible with RBBB diagnostics. Furthermore, ==different elevations and variations, including notable upslope ST segments observed in leads V1 to V3, can further affirm the presence of bundle branch blocks without overt ischemic patterns.== Additional leads such as II, III, aVR, and aVF do not exhibit significant anomalies, aside from mild ST variations, which are consistent across these leads without inferring significant ischemia or infarction. The computed average heart rate is normal, around 73 bpm, ==with consistent RR intervals suggesting regular sinus rhythm.== Although the overall ECG demonstrates sinus rhythm, the broad QRS complexes and rightward axis deviation are hallmarks of a right bundle branch block. These observations, together with consistent measurements, fully support the diagnosis of RBBB. No acute ischemic changes are discerned based on ST segment and T wave morphology across most of the ECG leads, correlating with the absence of acute myocardial infarction signs. The ECG also appears abnormal due to the prolonged QRS complex duration, supporting the finding of a RBBB when considered in conjunction with the specific electrocardiographic patterns observed.

**Ground-truth Report:** Sinus rhythm; Right bundle branch block; Abnormal ECG.

**PULSE Interpretation:** The electrocardiogram (ECG) reveals a sinus tachycardia, characterized by a rapid heart rate with a normal P-wave axis, indicating that the sinus node is functioning normally but at an accelerated pace. Additionally, the ECG shows a right bundle branch block (RBBB), which is a conduction abnormality affecting the right ventricle, resulting in a delayed activation of the right ventricle. Overall, the ECG suggests a diagnosis of sinus tachycardia with right bundle branch block.

Figure 6: Cardiologist Evaluations. Blue: Findings exceeding expert expectations. Yellow: Findings with differing expert opinions.

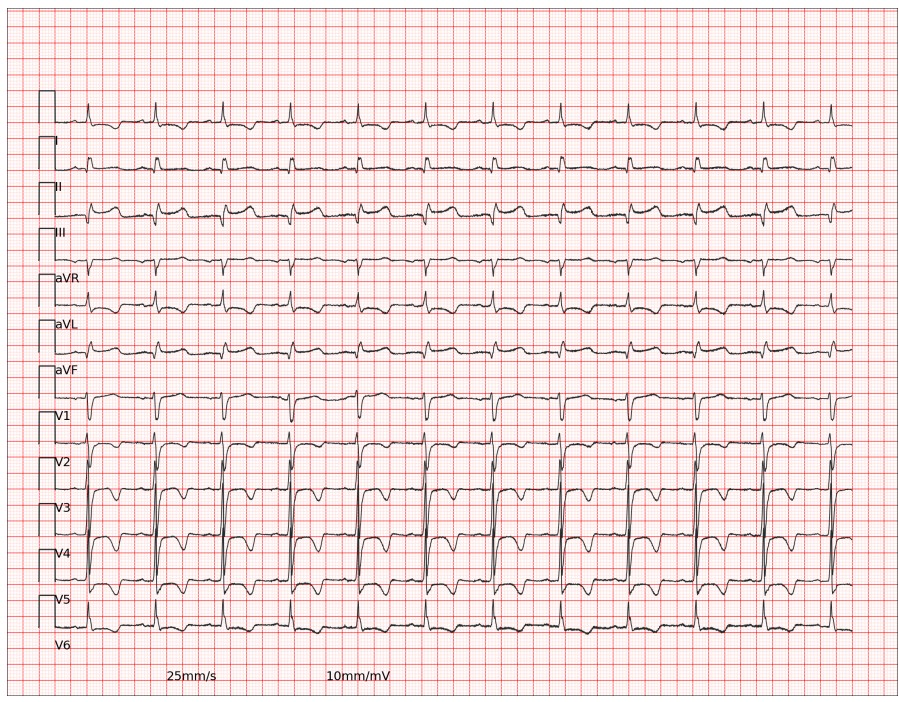

**GEM Interpretation:** Upon analyzing the provided ECG image and computed measurements, several key features and abnormalities are evident. The rhythm is sinus, as indicated by the regular P waves preceding each QRS complex. However, the QT interval is prolonged across multiple leads, with QTc values exceeding 500 ms in several instances, suggesting a risk for arrhythmias. In leads II, III, and aVF, there are notable changes consistent with an inferior infarct, including abnormal Q waves and ST segment deviations. Additionally, leads V2 to V4 show ST segment elevation and T wave inversion, which are indicative of anterior wall ischemia or infarction. The lateral leads, particularly I, aVL, V5, and V6, exhibit ST-T changes, such as ST segment depression and T wave inversion, which may suggest lateral myocardial ischemia. The presence of these changes in multiple leads supports the diagnosis of myocardial ischemia. The combination of these findings, including the prolonged QT interval and the ST-T changes, suggests a complex cardiac condition involving both ischemic and infarct patterns. The ECG is abnormal, with evidence pointing towards an inferior infarct, anterior and lateral ischemia, and a prolonged QT interval, all of which require further clinical correlation and management.

**Target Answer (GPT-4o):** Analyzing the provided 12-lead ECG shows evidence of an inferior myocardial infarction, as evidenced by significant deviations in the leads primarily reflective of the inferior wall, such as leads II, III, and aVF, which exhibit abnormalities in ST segments and T wave morphology. The QRS complexes and ST segment changes suggest an old or acute inferior infarct. Additionally, ST-T changes are notable in the anterior leads (V2-V4), indicating anterior or anteroseptal myocardial ischemia. The presence of significant ST elevation or depression, coupled with T wave inversion in these leads, corroborates the potential for ischemia in the anterior regions of the heart. In the lateral leads (I, aVL, V5, and V6), the presence of ST segment and T wave changes could indicate lateral wall ischemia or infarction. Specifically, notable changes in ST form and T wave morphology support this suspicion, which aligns with the observed abnormalities. The heart rate, computed at an average of 70 beats per minute, appears normal as it shows regular RR intervals, supporting a sinus rhythm without apparent rhythm disturbances. The computed PR intervals and QRS durations in different leads suggest no significant atrioventricular or bundle branch blocks, except for possible localized conduction delays that contribute to the differential ECG patterns observed. Collectively, these ECG findings demonstrate a combination of inferior infarct with the potential for ischemia notable in both anterior and lateral regions. This pattern of changes, particularly the ST-T variations across multiple leads, supports the likelihood of broadened myocardial involvement. The diagnosis aligns with the classification of an abnormal ECG, confirming myocardial ischemia and infarction.

**Ground-truth Report:** Sinus rhythm; Inferior infarct; Ant/septal and lateral ST-T changes may be due to myocardial ischemia; Abnormal ECG.

**PULSE Interpretation:** The electrocardiogram (ECG) reveals a sinus rhythm, indicating a normal heart rate and rhythm. However, it also shows an intraventricular conduction defect, which suggests a delay in the electrical conduction system of the heart. Furthermore, the ECG indicates evidence of a previous inferior myocardial infarction, although the age of the infarct cannot be determined. Additionally, there are ST-T changes observed in the anterior, septal, and lateral leads, which may be indicative of myocardial ischemia. Overall, the ECG is abnormal, suggesting cardiac disease. Clinical Diagnosis: Abnormal ECG with evidence of previous inferior myocardial infarction and possible myocardial ischemia.

Figure 7: Cardiologist Evaluations. Blue: Findings exceeding expert expectations. Yellow: Findings with differing expert opinions.

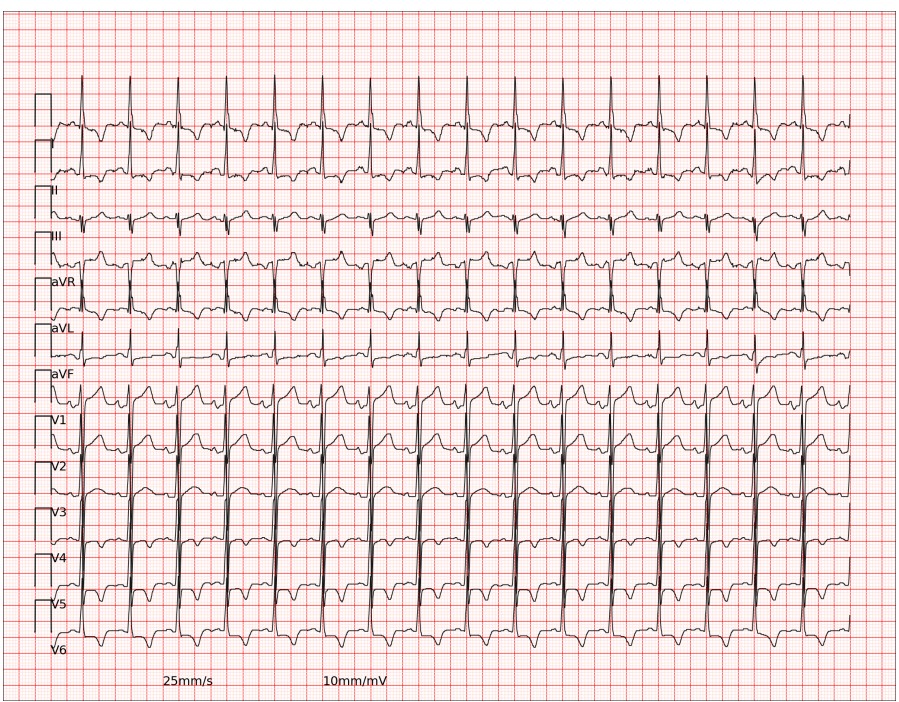

25mm/s        10mm/mV

**Target Answer (GPT-4o):** In analyzing the ECG image, several key features and abnormalities are identified that support the clinical findings. The rhythm across leads is sinus, as observed from the consistent P wave morphology, notably prominent in Lead II, indicating a regular atrial rhythm. In Lead II, the P wave amplitude varies, but the durations range around 92 to 104 ms, suggesting left atrial enlargement. This enlargement is further evidenced by the increased P wave amplitude in several leads, including V1, where the amplitude is significantly high. The QRS complexes show increased amplitudes across multiple leads, particularly in V3 and V4, with corresponding secondary ST-T changes. This pattern of high voltage QRS complexes and secondary repolarization abnormalities is indicative of left ventricular hypertrophy (LVH). The computer measurements confirm increased QRS amplitude in these leads, supporting the presence of LVH. Additionally, there is anterior ST segment elevation seen in leads V2-V4, which can be attributed to the strain pattern associated with LVH rather than acute myocardial infarction, as it correlates with the presence of LVH. There is no significant Q wave development or evidence of acute ischemic injury that suggests an infraction. Overall, the ECG findings collectively indicate sinus rhythm with left atrial enlargement and left ventricular hypertrophy with secondary repolarization abnormalities. The anterior ST elevation observed is likely a consequence of LVH, consistent with a LVH strain pattern. These interpretations align with the clinical manifestations of a patient with left heart strain and structural changes, corroborated by the data synthesis from both visual ECG interpretation and detailed interval measurements.

**Ground-truth Report:** Sinus rhythm; Left atrial enlargement; LVH with secondary repolarization abnormality; Anterior ST elevation, probably due to LVH.

Figure 8: Cardiologist Evaluations. Blue: Findings exceeding expert expectations. Yellow: Findings with differing expert opinions.

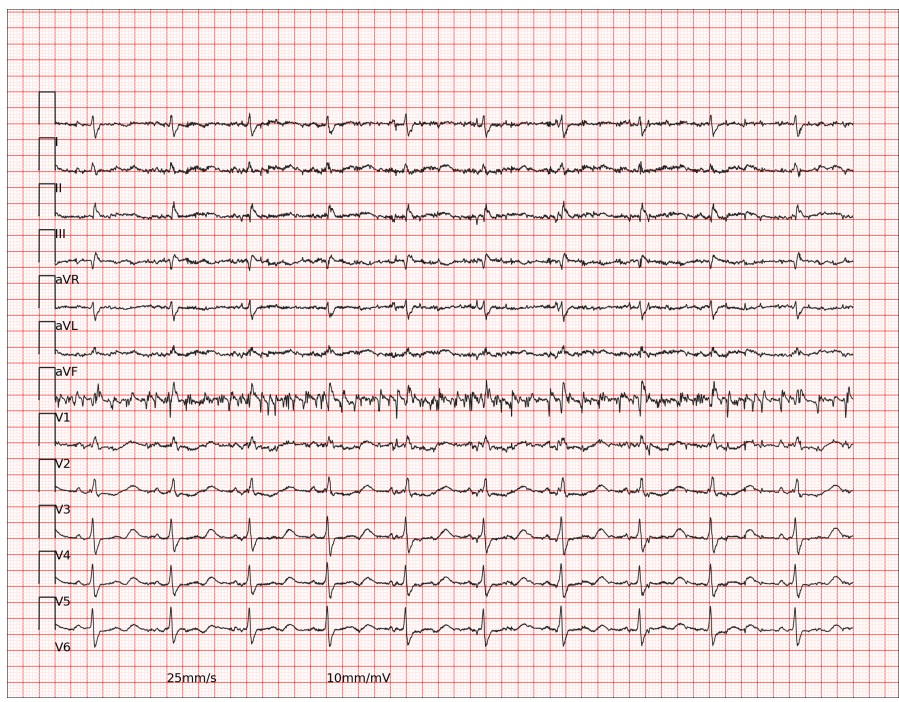

**GEM Interpretation:** Upon analyzing the provided ECG image and computed measurements, several key features and abnormalities are evident. The rhythm is sinus, as indicated by the consistent P wave morphology across leads, but there is a borderline first-degree A-V block, suggested by the prolonged PR intervals, particularly noticeable in leads I and II. The QRS complexes exhibit a right bundle branch block pattern, characterized by a widened QRS duration, particularly in lead V1, where an rsR' pattern is observed. The axis is rightward, which is consistent with the right bundle branch block. In the inferior leads (II, III, and aVF), there are nonspecific T wave changes, which could indicate ischemia or other non-specific changes. The QRS amplitude in lead III is notably low, which might suggest an inferior wall issue, although the changes are not definitive for infarction. The lateral leads (I, aVL, V5, and V6) show some T wave changes, but these are nonspecific and do not clearly indicate lateral ischemia or infarction. The QTc intervals are prolonged in several leads, which could be a concern for potential arrhythmogenic risk. The overall ECG is abnormal, with the combination of these findings suggesting a complex cardiac conduction issue. The rightward axis and right bundle branch block are significant, and the nonspecific T wave changes in the inferior and lateral leads warrant further clinical correlation and possibly additional diagnostic testing to rule out underlying ischemic heart disease or other cardiac conditions.

**Target Answer (GPT-4o):** Upon reviewing the 12-lead ECG image, the diagnosis reveals a sinus rhythm with a rightward axis and the presence of a right bundle branch block. The rhythm is consistent with a heart rate around 70 bpm, indicated by relatively regular RR intervals. The rightward axis is evident as lead I shows a smaller QRS amplitude compared to lead aVF, which reflects more positive deflections. A right bundle branch block is supported by the widened QRS complexes, particularly evident in leads V1 and V2, where an rsR' pattern could be observed, consistent with this conduction abnormality. Examination of the lateral leads (I, aVL, V5, V6) reveals mild T wave abnormalities, which are noted as nonspecific changes. In these leads, the morphology of the T waves shows slight variations that do not definitively point to ischemia but suggest some lateral wall changes. Despite these T wave changes being nonspecific, they may warrant further clinical correlation given the right bundle branch block already present. In summary, the ECG findings primarily indicate a right bundle branch block with a rightward axis. The T wave changes noted in the lateral leads are nonspecific and should be clinically correlated to assess their significance. These findings characterize the ECG as abnormal, and the conduction delay could suggest underlying pathology that may require additional testing or clinical evaluation to rule out underlying cardiac conditions. This evaluation is critical to ensure comprehensive cardiovascular assessment and appropriate subsequent management.

**Ground-truth Report:** Sinus rhythm; Rightward axis; Right bundle branch block; Lateral T wave changes are nonspecific; Abnormal ECG.

**PULSE Interpretation:** The electrocardiogram (ECG) reveals a sinus rhythm, indicating a normal heart rate and rhythm. However, the ECG also exhibits several abnormalities, including a prolonged QT interval, which may indicate a prolonged repolarization phase, and a right axis deviation, suggesting an abnormal orientation of the heart's electrical axis. Additionally, the ECG shows signs of right ventricular hypertrophy, low QRS voltages in the limb leads, and extensive ST-T changes, which may be attributed to either hypertrophy or ischemia, leading to an overall abnormal ECG.

Figure 9: Cardiologist Evaluations. Blue: Findings exceeding expert expectations. Yellow: Findings with differing expert opinions.

## A.6 Broader Impacts and Limitations

**Broader Impacts.** This work explores the potential of leveraging large language models to generate high-granularity ECG interpretations without the need for manual annotation. It contributes to the medical AI community in three key aspects: First, GEM is the first multimodal large language model that unifies time series, images, and text to enable feature-grounded analysis, evidence-based diagnosis, and clinician-like diagnostic workflows. This opens new avenues for research on multimodal alignment in ECG interpretation. Second, the proposed data generation methodology and the resulting ECG-Grounding dataset establish the first high-resolution resource for grounded ECG understanding. This dataset can support future research aimed at developing clinically applicable conversational ECG models. Third, the introduction of the Grounded ECG Understanding task provides a comprehensive and fine-grained evaluation protocol, helping future models to be assessed more thoroughly and precisely in terms of clinical reasoning and interpretability. By bridging computational precision with clinician-level reasoning, GEM represents a step toward more reliable, explainable, and clinically applicable AI-assisted ECG analysis, with potential for broader integration into real-world healthcare workflows.

On the other hand, it is important to acknowledge that the techniques explored in this study are intended for research purposes. Therefore, the proposed GEM model should not be directly used to make critical clinical decisions. Despite its strong performance in ECG interpretation, GEM is designed to serve as an assistive tool rather than a standalone solution for high-stakes clinical use.

**Limitations.** As discussed in Section 4.4, although our knowledge-guided instruction data generation approach avoids costly expert annotations while producing high-quality target answers, GPT-4o still occasionally generates responses that may not fully align with cardiologist interpretations. This limitation could be addressed by incorporating expert feedback into the data generation loop, which we identify as an important direction for future work.

