# OpenReview forum: "GEM: Empowering MLLM for Grounded ECG Understanding with Time Series and Images"
_NeurIPS.cc/2025/Conference — NeurIPS 2025 poster_

### Official Review · Reviewer_KnZY · 2025-07-02

**Clarity:** 3
**Significance:** 3
**Originality:** 3
**Rating:** 5
**Confidence:** 4

**Summary:**

This paper introduces a multimodal LLM for ECG interpretation that supports feature-based analysis, evidence-based reasoning, and a clinician-like diagnostic workflow. Across several datasets, the method outperforms prior PULSE-based models and includes a cardiologist evaluation.

**Questions:**

1. Generalisability beyond ECG

    Question. Could you either (i) add experiments on at least one additional medical modality (e.g., phonocardiograms, chest-x-ray images, or tabular vitals) or (ii) provide a principled analysis of why your architecture would—or would not—transfer to other signal types?

    Actionable guidance. A small-scale ablation (≤1 additional dataset) or a synthetic test showing that the model can ingest non-ECG inputs would be sufficient.

2. Dependence on GPT-4o-generated training data

    Question. What is the quantitative effect of replacing GPT-4o with (a) a weaker open-source LLM or (b) clinician-curated annotations for the same prompts?

     Actionable guidance. Please report performance when the synthetic corpus is produced by another model (e.g., MedAlpaca, or GPT-3.5) and, if feasible, when a small fraction is manually verified by cardiologists.

3. Validation of GPT-4o's clinical reasoning depth

    Question. Have you benchmarked GPT-4o on tasks closer to specialist reasoning (e.g., Board-style ECG interpretation questions with free-text answers) rather than multiple-choice USMLE items?

    Actionable guidance. Reporting such a benchmark or summarising known literature—and clarifying how any shortcomings might affect label quality—would help contextualise the training pipeline's reliability.

**Ethical Concerns:**

["NO or VERY MINOR ethics concerns only"]

**Final Justification:**

The authors have adequately addressed the concerns raised in the initial review.
The clarifications provided, particularly regarding generalization beyond ECG and dependence on GPT-4o generated training data, strengthen the manuscript significantly.

**Limitations:**

The authors should address the following critical limitation that appears to be missing from their current discussion:
Knowledge inheritance and error propagation: If there are errors in GPT-4o's knowledge base, there is a possibility that these same errors may also be present in this model. This represents a significant limitation as any systematic biases, factual inaccuracies, or knowledge gaps from the source model could be perpetuated or even amplified in the current system. The authors should explicitly acknowledge this risk and discuss how they have attempted to identify, validate, or mitigate such inherited errors.

**Quality:**

3

**Strengths And Weaknesses:**

### Strengths

1. Evaluation on multiple datasets shows the proposed approach consistently exceeds baseline methods.

2. Qualitative assessment by cardiologists corroborates the model's effectiveness.

3. The prompts used are provided, enhancing reproducibility.

### Weaknesses

1. The work is confined to ECG data; it lacks experiments or discussion on extending the method to other modalities or on the limitations imposed by this narrow focus. While this may suffice for a medical-domain venue, the broader audience of the present conference likely expects such analysis.

2. GPT-4o is employed to create training data, yet its medical knowledge has not been evaluated here. Although GPT-4o scores highly on the USMLE—multiple-choice questions aimed at licensure—this study tackles tasks involving specialist reasoning, and the gap is non-trivial. An empirical assessment or a discussion in the limitations section is needed. It would also help to report how performance changes when data are generated and evaluated with models other than GPT-4o.

---

> ### Author Rebuttal · Authors · 2025-07-30
>
> Thank you very much for your thoughtful comments! We are glad you recognized the value of our method and appreciated the open-source contributions in our work. We respond to your comments in detail below.
>
> >### **[Weakness 1, Question 1] Generalization beyond ECG**
>
> We appreciate the reviewer’s suggestion to consider generalization beyond ECG. While our work is intentionally focused on ECG, a modality that is not only foundational in cardiovascular diagnostics but also highly accessible and widely adopted across care settings, we agree that exploring generalizability is valuable.
>
> Following your suggestion, we conduct experiments extending our methods to respiratory spirogram data for pulmonary disease detection. The dataset was extracted from UK Biobank field 3066. As shown in the Table 1 below, our approach outperforms both deep learning SOTA and LLaMA3.1-8B across multiple metrics. These results suggest that our methods are not inherently limited to ECG and hold promise for broader clinical modalities.
>
> While these results demonstrate the potential generalizability of our framework, this paper is intentionally scoped to ECG interpretation. All modeling components and instruction data were designed around the specific structure and diagnostic reasoning challenges associated with ECG data. We believe ECG offers a rich and clinically meaningful testbed for studying multimodal reasoning in depth.
>
> In the revised version, we will add a discussion on how our framework could be adapted to other medical modalities. This primarily involves adjusting the knowledge-guided instruction data generation pipeline to account for modality-specific representations and clinical reasoning structures.
>
> ***Table 1: Experiment results on respiratory spirogram data for pulmonary disease detection***
>
> | Model              | AUROC  | AUPRC  | Sensitivity  | Specificity  | F1 score |
> |--------------------|--------|--------|--------------|--------------|----------|
> | Deep Learning SOTA | 82.7   | 80.7   | 69.1         | 79.0         | 70.1     |
> | Llama3.1-8B        | 76.9   | 73.2   | 98.4         | 12.5         | 64.2     |
> | Ours               | 89.8   | 90.5   | 83.3         | 67.0         | 74.4     |
>
>
> >### **[Weakness 2, Question 2] Dependence on GPT-4o generated training data**
>
> Thank you for raising this important point.
>
> To fully address your concern, we conduct a structured human evaluation covering three sources of outputs: GPT-4o generated data, Deepseek-R1 generated data, and GEM generated interpretations.
>
> In total, 400 ECG-Grounding data (200 from GPT-4o and 200 from Deepseek-R1) and 200 GEM's interpretation were independently reviewed by eight board-certified cardiologists, using seven predefined clinical criteria designed to assess both reliability and usefulness. This unified evaluation protocol allows us to (1) verify the quality of GPT-4o generated training data, (2) test the effectiveness of open-source substitutes, and (3) confirm the clinical utility of the GEM model.
>
> We report the evaluation results in the Table 2 below. The expert evaluation shows that GPT-4o consistently achieves high scores across both reliability and clinical usefulness, with particularly strong performance in analytical completeness and reasoning quality. These results demonstrate that, when using our knowledge-guided instruction data generation, GPT-4o is capable of generating high-quality ECG interpretations that are both clinically reliable and practically valuable.
>
> Deepseek-R1 is also capable of generating clinically acceptable, high-quality ECG interpretations with our methods. This demonstrates that our method is adaptable to alternative LLM backbones and remains applicable in settings without commercial API access. We will include these results in the revised version of the paper.
>
> Due to time and resource constraints, we have not yet completed GEM training and evaluation on Deepseek-R1 generated data. We plan to include this experiment in the revised paper to quantify the performance changes.
>
> ***Table 2: Human Evaluation on GPT-4o/Deepseek-R1 Generated Training Data and GEM's Interpretation***
>
> **Reliability metrics**
> |                | Analytical Relevance | Analytical Accuracy  | Analytical Completeness  |
> |----------------|----------------------|----------------------|--------------------------|
> | **GPT-4o**     | 4.7/5                | 4.6/5                | 4.7/5                    |
> | **Deepseek-R1**| 4.8/5                | 4.7/5                | 4.9/5                    |
> | **GEM**        | 4.6/5                | 4.4/5                | 4.6/5                    |
>
> **Usefulness metrics**
> |                | Reasoning Quality | Findings Novelty | Clinical Value | Overall Satisfaction |
> |----------------|-------------------|------------------|----------------|----------------------|
> | **GPT-4o**     | 4.7/5             | 4.4/5            | 4.7/5          | 4.5/5                |
> | **Deepseek-R1**| 4.8/5             | 4.5/5            | 4.6/5          | 4.7/5                |
> | **GEM**        | 4.6/5             | 3.9/5            | 4.3/5          | 4.4/5                |
>
> ```
> Scoring criteria (Note: Due to space limitations, we only present scores 4–5 here. The full scoring criteria (including 1–3) will be provided in the updated version):
>
> Analytical Relevance: Do the model’s analyses closely support the diagnosis, and is there corresponding ECG evidence?
> 5: Every analysis point is highly relevant to the diagnosis, with clear supporting evidence.
> 4: Most analyses are strongly relevant, with minor insufficiencies.
>
> Analytical Accuracy: Are there any medical factual errors in the model's output?
> 5: Completely accurate
> 4: Mostly accurate
>
> Analytical Completeness: Does the model comprehensively discuss key ECG components relevant to the diagnosis, including rhythm, intervals, and waveforms?
> 5: All relevant ECG features (rhythm, PR, QRS, ST, T waves, intervals, etc.) are accurately discussed.
> 4: Most key ECG features are covered, with minor omissions.
>
> Reasoning Quality: Does the model provide a clear, evidence-based reasoning process similar to that of a clinician, logically deriving the diagnosis from ECG features?
> 5: Clear and coherent reasoning structure, explaining each step from ECG to diagnosis causally.
> 4: Overall reasonable reasoning, but some steps lack detail.
>
> Findings Novelty: Does the model provide insights or findings not noticed by the clinician?
> 5: Important new diagnoses or findings.
> 4: Novel and somewhat insightful content.
> 3: Some new findings, but of limited value.
>
> Clinical Value: Does the model output help in clinical decision-making?
> 5: Direct and significant support for clinical judgment; content is clear and reliable.
> 4: Most content is helpful and practically useful.
>
> Overall Satisfaction: Subjective rating of the overall quality of this analysis.
> 5: Very satisfied.
> 4: Satisfied
> ```
>
>
> >### **[Question 3] Validation of GPT-4o's clinical reasoning depth**
>
> Directly using GPT-4o for board-style ECG interpretation tasks that require specialist-level reasoning typically yields suboptimal results. As shown in Table 3 of our paper, when prompted to generate free-text clinical reports on the PTB-XL dataset, GPT-4o achieves a report score of only 50.2, significantly lower than our method (67.1). This task requiring comprehensive interpretation and is close to to board-style questions. Moreover, GPT-4o also performs poorly on multiple-choice diagnostic tasks (as shown in main paper Table 2 ), highlighting its difficulty in making correct diagnosis even under simplified settings. This further underscores the challenge it faces in generating accurate and complete reasoning in free-text ECG interpretation.
>
> This is the reason we designed the knowledge-guided instruction data generation, where the GPT-4o is used as a controlled generation tool to construct training data, guided by ground-truth diagnostic reports and heartbeat-level descriptions. With our method, GPT-4o is able to generate high-quality reasoning data. This is supported by human evaluation on GPT-4o outputs, where cardiologists judged the reasoning to be clinically acceptable in most cases.
>
>
> >### **[Limitation 1]  Knowledge inheritance and error propagation risk**
>
> We appreciate the reviewer’s insightful comment regarding the risk of inherited errors from GPT-4o.
>
> We agree that if GPT-4o’s factual understanding or clinical knowledge is flawed, these errors could propagate into the generated training data. To mitigate this risk, we designed our data generation pipeline to anchor GPT-4o’s outputs to ground-truth clinical labels. Specifically, GPT-4o is prompted to explain or elaborate on known diagnoses and lead-level features from expert-labeled datasets such as MIMIC-IV-ECG. This constrains the model to generate reasoning that reflects verified clinical diagnosis, rather than freely recalling potentially unreliable knowledge.
>
> Our human evaluation on both GEM’s interpretation and GPT-4o generated training data verified that our data generation pipeline help substantially mitigate the risk of major factual inaccuracies. In future work, we plan to integrate clinical validation tools and expert feedback loops to further improve the reliability and safety of model-generated content.
>
> We will explicitly include this discussion in the revised manuscript.

---

> ### Comment · Reviewer_KnZY · 2025-08-04
> **Comments on rebuttal**
>
> Dear Authors,
>
> Thank you for the thorough and constructive rebuttal. I appreciate the additional experiments and clarifications. Below I summarize my current assessment and list a few points where further detail would strengthen the revision.
>
> # 1. Generalizability beyond ECG
>
> ### Commendation
> Your new spirogram experiment is a valuable addition and demonstrates that the proposed architecture is not intrinsically tied to ECG signals.
>
> ### Clarification requests
>
> 1. Training protocol – Was the same ECG‑trained model fine‑tuned on spirograms, or did you train a fresh model from scratch using the spirogram corpus?
>
> 2. Data scale and diversity – How many spirogram records and diagnostic labels were used, and how does their clinical and linguistic diversity compare with your ECG datasets (e.g., number of unique diagnoses, class balance, range of signal morphologies)?
>
> 3. Prompt / instruction adaptation – Did you modify the knowledge‑guided instruction templates for spirograms? If so, please provide two or three concrete examples.
>
> 4. Interpretation of results – Given that spirogram signals typically exhibit lower morphological variability than ECGs, could you elaborate on what aspects of the architecture or prompting strategy you believe enabled the observed performance gains?
>
> Including these details (even in an appendix) will help readers reproduce and interpret the generalization experiment.
>
> # 2. Dependence on GPT‑4o‑generated data
> Your human evaluation comparing GPT‑4o and DeepSeek‑R1 outputs is welcome. To maximise reproducibility and future benchmarking:
>
> - Would it be feasible to release the DeepSeek‑R1‑generated corpus and your cardiologist‑graded labels? Even a de‑identified subset would allow independent verification.
>
> - For transparency, please report inter‑rater reliability statistics (e.g., Fleiss’ κ) for the eight cardiologists across the seven clinical criteria.
>
> # 3. Validation of GPT‑4o’s clinical reasoning
> Your strategy of anchoring GPT‑4o outputs to ground‑truth labels is sensible. Have you catalogued any systematic errors that still leak through? Even a short qualitative summary would inform practitioners about residual risks.
>
> Overall, the rebuttal addresses the majority of concerns, and I am encouraged by the additional evidence supplied. Clarifying the points above will, in my view, make the final manuscript even more impactful and reproducible.

---

> > ### Author Response · Authors · 2025-08-07
> >
> > Dear Reviewer KnZY,
> >
> > Given that the discussion deadline is approaching, we would like to kindly check whether our responses have addressed your concerns. We are happy to continue the discussion, and if you have any remaining questions, please feel free to raise them. We will do our best to address your concerns until the end of the discussion period.
> >
> > Thank you!
> >
> > Authors

---

> > > ### Comment · Reviewer_KnZY · 2025-08-07
> > > **Reply to authors.**
> > >
> > > Thank you for your comprehensive response to the reviewers' comments. The authors have adequately addressed the concerns raised in the initial review. Based on the responses provided, I am updating my recommendation for this manuscript.

---

> ### Author Response · Authors · 2025-08-05
> **Reply to reviewer's comments (Part 1)**
>
> Thank you for your reply! Here are our clarifications on the points you raised.
>
> ## Generalizability beyond ECG
>
> > ### **Training protocol**
>
> We trained a fresh model (i.e., no prior domain-specific training) from scratch using the spirogram corpus to ensure a meaningful and modality-specific evaluation. Directly applying the ECG-trained model to spirograms can result in suboptimal and uninterpretable performance due to domain-specific feature dependencies and differences in signal morphology. Training from scratch ensures that the model learns spirogram-specific representations without being biased by ECG-specific priors. Moreover, in practical clinical applications, it is uncommon to deploy a single model to simultaneously interpret both ECGs and spirograms, as each modality requires distinct input structures, clinical features, and reasoning frameworks.
>
> >### **Data scale and diversity**
>
> We trained on a cohort of 18,416 patients selected from the UK Biobank (field 3066). The target task is the diagnosis of Chronic Obstructive Pulmonary Disease (COPD), formulated as a binary classification task. The dataset's diagnostic labels are categorized into two groups: confirmed COPD (8,245 patients) and non-COPD (10,171 non-COPD controls). In contrast, our ECG-based tasks are multi-label classification tasks covering a broader set of unique diagnosis.
>
> Compared to ECGs, which consist of high-frequency, multi-phase waveforms with distinct temporal segments (e.g., P wave, QRS complex, T wave), spirogram signals typically exhibit lower-frequency, continuous curves such as flow-volume or volume-time curves. While spirograms appear morphologically smoother, their diagnostic interpretation still requires reasoning over shape characteristics (e.g., concavity, slope changes) and integration of quantitative measurements like FEV1 and FVC.
>
> In this sense, the morphological variability is more subtle but still clinically meaningful, and effective interpretation depends on combining structural features with domain-specific diagnostic logic.
>
> >### **Prompt / instruction adaptation: Did you modify the knowledge‑guided instruction templates for spirograms?**
>
> Yes, adapting the knowledge-guided instruction templates to fit the spirogram modality is crucial, as it ensures that the prompts capture modality-specific features and align with the clinical reasoning process for diagnostics.
>
> The overall prompting pipeline remains consistent:
>
> (1) we first extract a morphological description from the spirogram curve (similar to ECG feature extraction).
>
> (2) we incorporate relevant patient-level metadata such as pulmonary function test (PFT) values.
>
> (3) we use the knowledge from the GOLD guidelines to guide the LLM in generating final, high-quality training data.
>
> Below are two examples of the generated training data:
>
>     case 1 (non-copd):
>         The patient is a 44-year-old female with no history of smoking, presenting for pulmonary function testing. Spirometry results reveal an FEV1 of 2.82 L (91.28% predicted) and an FVC of 3.62 L (94.62% predicted). The post-bronchodilator FEV1/FVC ratio is 0.779, which is above the diagnostic threshold of 0.70 for airflow obstruction as per GOLD criteria. Additionally, neither the FEV1 nor the FVC falls below the lower limit of normal (LLN), with FEV1 at -0.694 z-score and FVC at -0.422 z-score. The FEF25-75 is 2.95 L/s (94.51% predicted), also within normal limits. The spirometry graph description demonstrates a rapid initial rise, a rounded peak expiratory flow, and a gradually descending limb with mild concavity, consistent with normal expiratory flow patterns without significant obstruction. Given these findings, there is no evidence of fixed airflow obstruction. The absence of smoking history further reduces the likelihood of COPD. Therefore, the diagnostic criteria for COPD are not met based on the current pulmonary function testing and clinical context.
>
>     case 2 (copd):
>         The patient is a 62-year-old male with a significant smoking history, presenting for pulmonary function testing. Spirometry reveals a post-bronchodilator FEV1/FVC ratio of 0.61, which is below the diagnostic threshold of 0.70 as established by GOLD criteria for airflow obstruction. The FEV1 is 74.866% of predicted, indicating moderate airflow limitation. The FVC is within normal limits at 94.411% of predicted. The FEF25-75, a marker of small airway function, is reduced at 55.38% of predicted, further supporting obstructive physiology. The spirometry curve demonstrates an obstructive pattern with a steep initial rise to peak flow followed by a concave descending limb, consistent with airflow limitation. Given the patient's age, smoking history, and spirometric evidence of fixed airflow obstruction not fully reversible with bronchodilators, the diagnosis of COPD is confirmed.

---

> ### Author Response · Authors · 2025-08-05
> **Reply to reviewer's comments (Part 2)**
>
> >### **Interpretation of results**
>
> We believe that two factors collectively contribute to the observed performance gains:
>
> (1) Reliable extraction of core morphological features:
>
> In Table 1, our method employs the Deep Learning SOTA as the encoder to capture morphological features from the spirogram curves, and uses Llama 3.1-8B as the base LLM to process the complementary textual inputs.
>
> The comparison between the Deep Learning SOTA and Llama3.1-8B highlights the importance of robust morphological feature extraction for accurate spirogram interpretation.
>
> (2) Incorporation of domain-specific diagnostic logic aligned with clinical guidelines:
>
> The comparison between the Deep Learning SOTA and the full model demonstrates the importance of incorporating domain-specific diagnostic logic aligned with established clinical guidelines (i.e., knowledge-guided instruction data generation).
>
> These results further confirm the generalizability and robustness of our approach.
>
>
> ## Dependence on GPT‑4o‑generated data
>
> >### **Would it be feasible to release the DeepSeek‑R1‑generated corpus and your cardiologist‑graded labels?**
>
> Yes, we will release these data after appropriate de-identification.
>
> >### **Inter‑rater reliability statistics.**
>
> We appreciate the suggestion to report inter-rater reliability to further imporve the transparency. While Fleiss’ κ is commonly used, it requires that (1) each item is rated by the same number of raters, and (2) the ratings are nominal.
>
> In our human evaluation, each cardiologist assessed a different subset of the 200 cases from a given model (i.e., not all items were rated by the same raters). Additionally, the seven criteria were scored on ordinal 5-point scales rather than categorical labels. Therefore, Fleiss’ κ is not applicable in current setting.
>
> Instead, to ensure transparency, we report in Table 3 the mean and standard deviation of scores across all cardiologists for each clinical criterion.
>
> The mean reflects the overall evaluation of the model's outputs, and the standard deviation serves as a proxy for inter-rater variability (i.e., higher std indicates greater variability across raters). The standard deviation will be added in our revision.
>
> We plan to include the inter-rater reliability statistics such as Krippendorff’s α in future large-scale evaluations where more overlapping annotations are available.
>
> ***Table 3: Human Evaluation on Generated Data and GEM's Interpretation (Mean and STD)***
>
> **Reliability metrics**
> |                | Analytical Relevance | Analytical Accuracy  | Analytical Completeness  |
> |----------------|----------------------|----------------------|--------------------------|
> | **GPT-4o**     | 4.7/5 (0.66)         | 4.6/5 (0.82)         | 4.7/5 (0.65)             |
> | **Deepseek-R1**| 4.8/5 (0.57)         | 4.7/5 (0.78)         | 4.9/5 (0.42)             |
> | **GEM**        | 4.6/5 (0.60)         | 4.4/5 (0.80)         | 4.6/5 (0.57)             |
>
> **Usefulness metrics**
> |                | Reasoning Quality | Findings Novelty | Clinical Value | Overall Satisfaction |
> |----------------|-------------------|------------------|----------------|----------------------|
> | **GPT-4o**     | 4.7/5 (0.67)      | 4.4/5 (1.18)     | 4.7/5 (0.73)   | 4.5/5 (0.87)         |
> | **Deepseek-R1**| 4.8/5 (0.62)      | 4.5/5 (0.91)     | 4.6/5 (0.82)   | 4.7/5 (0.77)         |
> | **GEM**        | 4.6/5 (0.64)      | 3.9/5 (1.25)     | 4.3/5 (0.89)   | 4.4/5 (0.82)         |
>
> ##  Validation of GPT‑4o’s clinical reasoning
>
> >### **Have you catalogued any systematic errors that still leak through?**
>
> So far, we have not observed any systematic errors in GPT-4o outputs that lead to severe factual inaccuracies or propagate into GEM’s training data under our data generation pipeline.
>
> As discussed in Section 4.4 of the main paper, the most notable deviation we have observed is occasional disagreement between GPT-4o explanations and cardiologist interpretations. These discrepancies may arise in finer-grained feature judgments, such as waveform morphology or interval thresholds.
>
> Such disagreements are not unexpected, as they reflect the inter-observer variability commonly seen in real-world clinical interpretation, even among experienced cardiologists. Importantly, GPT-4o explanations are explicitly conditioned on ground-truth diagnostic labels derived from original clinical reports, which themselves represent expert assessments made at the time of care. As such, GPT-4o outputs reflect one valid expert viewpoint, while the reviewing cardiologist may provide another.
>
> We consider this inter-observer variability inherent to clinical diagnostic tasks, especially in morphology-sensitive settings, rather than a failure of the generative process.

---

### Official Review · Reviewer_Zvng · 2025-07-02

**Clarity:** 3
**Significance:** 3
**Originality:** 3
**Rating:** 5
**Confidence:** 4

**Summary:**

This paper introduces GEM, a novel MLLM designed for grounded and interpretable ECG analysis. The authors address two key limitations in existing models: the insufficient synergy between ECG time series and image modalities, and the lack of explainability linking diagnoses to specific waveform evidence. To tackle these issues, GEM uniquely integrates a dual-encoder architecture for both time series and images, along with a cross-modal alignment mechanism to enable holistic understanding. A major contribution is the proposal of a knowledge-guided method for automatically generating ECG-Grounding, the first high-granularity instruction dataset that explicitly connects diagnoses to measurable physiological parameters. Furthermore, the paper introduces the Grounded ECG Understanding task, a new clinically-oriented benchmark designed to comprehensively evaluate the model's evidence-based reasoning capabilities.

**Questions:**

Q1: The Diagnosis Guider is a core component that relies on a predefined set of rules. Could you elaborate on its robustness? Specifically, how does this rule-based system handle rare or complex cardiac conditions that may not be explicitly covered by the existing rules, and what is its potential failure mode in such scenarios?

Q2: The experimental results show a counter-intuitive finding where the GEM variant based on the general-domain LLaVA performs on par with, or even slightly better than, the variant based on the ECG-pretrained PULSE. Could you provide a possible explanation for this? Does this suggest that the pre-training strategy of PULSE might introduce a certain bias that is suboptimal for learning the new evidence-based grounding task you propose?

**Ethical Concerns:**

["NO or VERY MINOR ethics concerns only"]

**Final Justification:**

My concerns have been well-addressed. The authors provided detailed explanations and supplementary experimental results that have resolved my questions. I believe my current rating is appropriate.

**Limitations:**

Yes

**Quality:**

3

**Strengths And Weaknesses:**

S1. The proposed GEM model is the first to synergistically unify ECG time series, images, and text, establishing a novel multi-modal framework for more comprehensive ECG interpretation.

S2. The creation of the ECG-Grounding dataset via a knowledge-guided generation process provides the high-granularity resource for training and evaluating evidence-based diagnostic models.

S3. The introduction of the Grounded ECG Understanding task offers a new, clinically-oriented benchmark that enables a more thorough assessment of a model's explainability and grounding capabilities beyond traditional metrics.

W1. The paper's parameter-efficient fine-tuning strategy, which freezes the encoders, may prevent the model from optimally adapting its feature extraction to the new grounding task. An exploration of full end-to-end fine-tuning is a notable omission.

W2. The quality of the auto-generated ECG-Grounding dataset is not quantitatively validated by human experts, which is a critical step to ensure its reliability for training and evaluation.

W3. The ablation study lacks an "IMG-only" baseline, making it difficult to fully assess the complementarity of the two modalities. Furthermore, the marginal performance gains from adding images in some tasks (Table 2) do not fully support the strong synergy claim.

W4. The paper lacks a systematic analysis of failure cases, making it difficult to pinpoint whether errors stem from faulty feature extraction or flawed logical reasoning within the model.

---

> ### Author Rebuttal · Authors · 2025-07-30
>
> We appreciate your recognition of the innovation and contributions in our work! We are glad that you found the paper novel and the studied problem is important. We address your concerns point by point as follows.
>
> >### **[Weakness 1] Parameter-efficient fine-tuning may limit model adapting to new grounding task**
>
> Thank you for raising this important point. We agree that unfreezing the encoders could potentially improve the model’s ability to adapt feature extraction to the grounding task, particularly in a multimodal setting where fine-grained cross-modal alignment is essential.
>
> However, fully training all encoders incurs substantial computational cost, as it requires training two additional Transformer encoders.
> For this reason, we designed a cross-modal alignment learning module and adopted a widely used parameter-efficient training strategy (e.g., as in LLaVA). This design choice significantly reduced training cost and memory usage while maintaining good performance.
>
> We appreciate this suggestion and see end-to-end fine-tuning as a valuable next step. We plan to include this experiment in the revised version to better understand its impact on grounding performance.
>
>
> >### **[Weakness 2] Data quality evaluation with human experts**
>
> We conduct a quantitative human evaluation on 200 unique ECG cases, randomly sampled from ECG-Grounding generated by GPT-4o.
>
> The evaluation is performed by eight board-certified cardiologists, each independently assessing a subset of cases using seven predefined criteria designed to measure both reliability and clinical usefulness.
>
> Table 1 summarizes the results. The expert evaluation shows that GPT-4o consistently achieves high scores across both reliability and clinical usefulness. These findings indicate that, under our knowledge-guided instruction data generation, GPT-4o can produce ECG interpretations that are both clinically reliable and practically valuable.
>
> ***Table 1: Human Evaluation on GPT-4o Generated Training Data***
>
> **Reliability metrics**
> |                | Analytical Relevance | Analytical Accuracy  | Analytical Completeness  |
> |----------------|----------------------|----------------------|--------------------------|
> | **GPT-4o**     | 4.7/5                | 4.6/5                | 4.7/5                    |
>
> **Usefulness metrics**
> |                | Reasoning Quality | Findings Novelty | Clinical Value | Overall Satisfaction |
> |----------------|-------------------|------------------|----------------|----------------------|
> | **GPT-4o**     | 4.7/5             | 4.4/5            | 4.7/5          | 4.5/5                |
>
>
> ```
> Scoring criteria (Note: Due to space limitations, we only present scores 4–5 here. The full scoring criteria (including 1–3) will be provided in the updated version):
>
> Analytical Relevance: Do the model’s analyses closely support the diagnosis, and is there corresponding ECG evidence?
> 5: Every analysis point is highly relevant to the diagnosis, with clear supporting evidence.
> 4: Most analyses are strongly relevant, with minor insufficiencies.
>
> Analytical Accuracy: Are there any medical factual errors in the model's output?
> 5: Completely accurate
> 4: Mostly accurate
>
> Analytical Completeness: Does the model comprehensively discuss key ECG components relevant to the diagnosis, including rhythm, intervals, and waveforms?
> 5: All relevant ECG features (rhythm, PR, QRS, ST, T waves, intervals, etc.) are accurately discussed.
> 4: Most key ECG features are covered, with minor omissions.
>
> Reasoning Quality: Does the model provide a clear, evidence-based reasoning process similar to that of a clinician, logically deriving the diagnosis from ECG features?
> 5: Clear and coherent reasoning structure, explaining each step from ECG to diagnosis causally.
> 4: Overall reasonable reasoning, but some steps lack detail.
>
> Findings Novelty: Does the model provide insights or findings not noticed by the clinician?
> 5: Important new diagnoses or findings.
> 4: Novel and somewhat insightful content.
> 3: Some new findings, but of limited value.
>
> Clinical Value: Does the model output help in clinical decision-making?
> 5: Direct and significant support for clinical judgment; content is clear and reliable.
> 4: Most content is helpful and practically useful.
>
> Overall Satisfaction: Subjective rating of the overall quality of this analysis.
> 5: Very satisfied.
> 4: Satisfied
> ```
>
> >### **[Weakness 3]  IMG-Only baseline and performance gain from adding images**
>
> Thank you for the insightful suggestion. We did not train a separate IMG-only variant of GEM since PULSE can be seen as the IMG-only baseline, as it shares the same configuration used in our ablation study.
>
> The only difference is that PULSE was trained for three epochs, whereas our ablation variants were trained for one epoch due to resource constraints. Despite this, the TS+IMG still outperforms PULSE in most tasks, suggesting that adding modalities improves both training efficiency and performance, and that the two modalities are complementary. We will clarify this point in the revised version.
>
> In addition to performance, we believe that adding the image modality also improves usability, for example, by making it easier for users to interact with the system using a photo of their ECG.
>
> >### **[Weakness 4] Failure case analysis**
>
> Thank you for this insightful and valuable comment. We conduct a systematic analysis of failure cases, particularly informed by the expert feedback collected during the human evaluation. Based on these observations, we categorize the major failure types and analyze potential causes as follows:
>
> * Incorrect diagnosis
>
>     These errors may result from limitations in the representation stage. For example, certain morphological features such as subtle ST-segment changes or P-wave abnormalities were sometimes missed by the encoders.
>
>     Potential causes include:
>
>     * Representation deficiency: Some features were not captured due to encoder limitations.
>
>     * Training data limitations: The current training set may not cover the full spectrum of ECG variations, especially when data distribution shifts.
>
>
> * Overstating the severity of findings
>
>     GEM occasionally exaggerates the severity of certain cardiac conditions, which some cardiologists noted could lead to unnecessary patient anxiety.
>
>     Potential causes include:
>
>     * Limited context on patient history: In real-world settings, physicians contextualize ECG findings using patient history, which GEM does not have access to, potentially leading to overdiagnosis in isolation.
>
> In the revised version, we will include this failure case analysis to provide a more systematic view of where errors arise. These insights will not only contextualize current performance but also inform future directions for improving model safety, reliability, and alignment with clinical practice.
>
> >### **[Question 1] Robustness of Diagnosis Guider and how to handle rare or complex cardiac conditions**
>
> Thank you for this important question regarding the robustness of the Diagnosis Guider. We address your concerns as follows:
>
> First, rule-based interpretation is the foundation of clinical ECG analysis in current medical practice. For example, the widely adopted AHA/ACCF/HRS guidlines [1] provide structured diagnostic criteria for arrhythmias, conduction delays, and ischemic changes. Commercial ECG systems such as GE MUSE [2] and Philips DXL [3] also rely on deterministic rule sets for generating diagnostic statements. Our Diagnosis Guider is developed with reference to standard clinical guidelines, aiming to analyze ECGs in a way aligned with cardiology practice.
>
> Second, we acknowledge that rule-based systems, like clinical interpretation itself, can be fallible, especially in rare disease or complex conditions. To improve the overall robustness, our framework does not rely solely on the Diagnosis Guider. It contains learning-based components that can capture broader data-driven patterns. This hybrid design enhances the overall robustness compared to standalone rule-based interpretation. Furthermore, our human evaluation shows that cardiologists found Diagnosis Guider outputs clinically reliable and useful. Together, these results indicate that the system is capable of handling a wide range of ECG interpretation scenarios with performance approaching clinical applicability.
>
> Third, failure modes may still arise, particularly for rare, under-represented, or novel cardiac conditions that fall outside both the current rule base and training data distribution. While we recognize that the system is not yet perfect, the overall performance, both in benchmark evaluations and in expert review, indicates that it has reached a level that helpful in many settings and can provide practical clinical value.
>
> In the future, we plan to further strengthen the Diagnosis Guider by expanding the rule set with expert input, incorporating more diverse training data, and iteratively refining the framework to improve its coverage and adaptability over time.
>
> [1] Rautaharju, Pentti M., Borys Surawicz, and Leonard S. Gettes. "AHA/ACCF/HRS recommendations for the standardization and interpretation of the electrocardiogram." Journal of the American College of Cardiology 53.11 (2009): 982-991.
>
> [2] GE HealthCare. (2019). Implementing MUSE to Streamline ECG Workflow Case Study.
>
> [3] Philips DXL ECG Algorithm Physician’s Guide.
>
> >### **[Question 2] Explanation for SFT LLaVA’s performance compared to SFT PULSE**
>
> This finding can be explained by the fact that PULSE was not pretrained on any high-granularity grounding data. As a result, PULSE lacks the necessary capabilities for grounded ECG interpretation. The knowledge embedded during its pretraining provides limited support for the evidence-based grounding task we propose.

---

> > ### Comment · Reviewer_Zvng · 2025-08-07
> >
> > Thank you for the detailed rebuttal and the additional effort invested in the work. The responses have addressed my concerns, and I believe my current scores appropriately reflect my positive assessment of the paper. I will maintain my scores.

---

### Official Review · Reviewer_W2KZ · 2025-07-03

**Clarity:** 3
**Significance:** 4
**Originality:** 3
**Rating:** 5
**Confidence:** 4

**Summary:**

The authors introduce GEM, an MLLM for 12-lead ECG interpretation that fuses raw time series, rendered ECG images, and text prompts. A dual-encoder architecture aligns signal and image embeddings, which are then used to prompt a GPT-4o model to generate high-granularity instructional data for the LLM. Instruction-tuning on a 30K samples ECG-Grounding dataset lets GEM generate cardiologist-style reports that explicitly refer to waveform evidence. Experiments on a new Grounded-ECG benchmark and on standard classification tasks show reasonable gains in diagnosis accuracy and explanation quality over prior works.

**Questions:**

1. Did the authors run experiments substituting GPT-4o with an open-source alternative? If yes, what kind of a performance dip can one expect?

2. Did the authors experiment with more LLM decoders? Did swapping those have a significant impact on performance, as observed in the case of JoLT (and BLIP-2)?

**Ethical Concerns:**

["NO or VERY MINOR ethics concerns only"]

**Final Justification:**

I really appreciate the work put the original submission and the rebuttal, and I appreciate the overall quality of the paper, and the depth of experiments performed. Based on a thorough review of the paper again, including evaluating the novelty embodied in the paper, and the depth of the experiments performed, I believe a rating of 5 is suitable for the paper. Thus, I am going to retain the original score. I applaud the authors for their systematic rebuttal and the overall quality of the paper.

**Limitations:**

Yes.

**Paper Formatting Concerns:**

No issues.

**Quality:**

3

**Strengths And Weaknesses:**

Strengths:
1. The paper is very well written, the task is well explained and motivated, and the GEM architecture is very well illustrated. The illustrations, in particular, make the paper easy to follow.

2. The authors have done fairly extensive evaluation on eight different metrics, and the ablations justify the use of time series and image input.

3. Strong reproducibility: The authors have made their code public, and the prompts and metrics are very well detailed in the appendix. The ECG-grounding dataset has also been made public.

4. Strong performance gains over Pulse and GPT-4o for both in-domain and out-of-domain ECG-Understanding, and ECG-Bench abnormality detection. Evidence-linked explanations further improve transparency, and the qualitative examples show substantially better explainability than baselines.

Weaknesses:

1. Owing to the utilization of GPT-4o in the pipeline, I would an estimate API cost for training GEM. Similarly, inference time numbers are missing.

2. The architecture, while novel, takes some inspiration from JoLT (Cai et al., 2023), which also does ECG interpretation, which isn't compared and contrasted with in the paper.

3. Although this has been duly noted by the authors, GPT-4o interpretations might occasionally not align with those by experts. This is more critical in the healthcare domain, which GEM is focused on, compared to other domains.

---

> ### Author Rebuttal · Authors · 2025-07-30
>
> Thank you very much for your encouraging comments on our paper! We are glad that you found that the paper is presented and articulated very clearly. We provide detailed responses to your concerns as follows.
>
> >### **[Weakness 1, Question 1] GPT-4o cost, inference time numbers, scalability**
>
> - GPT-4o cost and inference time numbers
>
>     Thank you for raising this important point. In our implementation, generating each sample required approximately 6,000 input tokens and 300 output tokens, resulting in a total cost of roughly USD $3,000 using the OpenAI GPT-4o API for the 300,000 samples in ECG-Grounding dataset. The average inference time per sample is 8s on a single A100 GPU.
>
> - Substituting GPT-4o with an open-source alternative
>
>     We test an open-source substitute model, the latest Deepseek-R1. We conduct a human evaluation on 400 ECG-Grounding data (200 from GPT-4o and 200 from Deepseek-R1).
>
>     The human evaluation is conducted by eight board-certified cardiologists, each independently evaluating a subset using seven predefined criteria that assess both reliability and usefulness in clinical practice.
>
>     As shown in the human evaluation results in Table 1, Deepseek-R1 is also capable of generating clinically acceptable, high-quality ECG interpretations with our methods. This demonstrates that our method is adaptable to alternative LLM backbones and remains applicable in settings without commercial API access. We will include these results in the revised version of the paper.
>
>     Due to time and resource constraints, we have not yet completed GEM training and evaluation on Deepseek-R1 generated data. We plan to include this experiment in the revised paper to quantify the performance changes.
>
>
>     ***Table 1: Human Evaluation on GPT-4o/Deepseek-R1 Generated Training Data***
>
>     **Reliability metrics**
>     |                | Analytical Relevance | Analytical Accuracy  | Analytical Completeness  |
>     |----------------|----------------------|----------------------|--------------------------|
>     | **GPT-4o**     | 4.7/5                | 4.6/5                | 4.7/5                    |
>     | **Deepseek-R1**| 4.8/5                | 4.7/5                | 4.9/5                    |
>
>     **Usefulness metrics**
>     |                | Reasoning Quality | Findings Novelty | Clinical Value | Overall Satisfaction |
>     |----------------|-------------------|------------------|----------------|----------------------|
>     | **GPT-4o**     | 4.7/5             | 4.4/5            | 4.7/5          | 4.5/5                |
>     | **Deepseek-R1**| 4.8/5             | 4.5/5            | 4.6/5          | 4.7/5                |
>
>     ```
>     Scoring criteria (Note: Due to space limitations, we only present scores 4–5 here. The full scoring criteria (including 1–3) will be provided in the updated version):
>
>     Analytical Relevance: Do the model’s analyses closely support the diagnosis, and is there corresponding ECG evidence?
>     5: Every analysis point is highly relevant to the diagnosis, with clear supporting evidence.
>     4: Most analyses are strongly relevant, with minor insufficiencies.
>
>     Analytical Accuracy: Are there any medical factual errors in the model's output?
>     5: Completely accurate
>     4: Mostly accurate
>
>     Analytical Completeness: Does the model comprehensively discuss key ECG components relevant to the diagnosis, including rhythm, intervals, and waveforms?
>     5: All relevant ECG features (rhythm, PR, QRS, ST, T waves, intervals, etc.) are accurately discussed.
>     4: Most key ECG features are covered, with minor omissions.
>
>     Reasoning Quality: Does the model provide a clear, evidence-based reasoning process similar to that of a clinician, logically deriving the diagnosis from ECG features?
>     5: Clear and coherent reasoning structure, explaining each step from ECG to diagnosis causally.
>     4: Overall reasonable reasoning, but some steps lack detail.
>
>     Findings Novelty: Does the model provide insights or findings not noticed by the clinician?
>     5: Important new diagnoses or findings.
>     4: Novel and somewhat insightful content.
>     3: Some new findings, but of limited value.
>
>     Clinical Value: Does the model output help in clinical decision-making?
>     5: Direct and significant support for clinical judgment; content is clear and reliable.
>     4: Most content is helpful and practically useful.
>
>     Overall Satisfaction: Subjective rating of the overall quality of this analysis.
>     5: Very satisfied.
>     4: Satisfied
>     ```
>
> >### **[Weakness 2] Discussion on JoLT**
>
> Thank you for bringing JoLT to our attention, we will include a detailed discussion of JoLT in the revised paper, highlighting similarities and differences in model architecture, supervision strategy, and output format.
>
> >### **[Weakness 3] Disagreement between GPT-4o and experts**
>
> We recognize that disagreement between GPT-4o-generated interpretations and expert opinions is particularly important in the healthcare domain, where model outputs may directly influence clinical reasoning. We interpret these disagreements not simply as model failures, but rather as reflections of the inherent variability in clinical ECG interpretation.
>
> Overall, the diagnostic conclusions are often aligned between models and human experts. However, discrepancies may arise in finer-grained feature judgments, such as waveform morphology or interval thresholds. This is not unexpected, it mirrors real-world clinical practice, where interpretation differences at the detail level are common even among experienced cardiologists.
>
> Importantly, GPT-4o generated explanations are conditioned on the ground-truth diagnosis from the original clinical reports, which represent a physician’s interpretation at the time of care. As a result, GPT-4o offers one valid expert perspective, while the reviewing cardiologist may offer another. This is consistent with known inter-observer variability, particularly in morphology-sensitive cases.
>
> In addition, expert evaluation in Table 1 confirms that GPT-4o generated ECG-Grounding data satisfy clinical requirements in the majority of cases. The model consistently receives high scores across both reliability and usefulness, with particularly strong performance in analytical completeness and reasoning quality. These results indicate that, under our knowledge-guided instruction data generation framework, GPT-4o can produce ECG interpretations that are not only clinically relevant but also aligned with the expectations of cardiologists.
>
> We also view these discrepancies as valuable opportunities: they highlight cases where model reasoning could be refined with human feedback, and they underscore the importance of modeling expert disagreement to better capture clinical uncertainty. In future work, we plan to explore multi-expert consensus frameworks, inspired by real-world clinical workflows for handling disagreements, such as multi-physician case reviews, to further improve model reliability and alignment with clinical standards.
>
> >### **[Question 2] Experiment with more LLM decoders**
>
> Thank you for the insightful question. We test GEM using Qwen2-7B as the LLM decoder, and the results on ECG-Bench are included in Table 2 below.
>
> Unlike the performance gains observed in JoLT or BLIP-2 with decoder swapping, we do not observe significant improvements when replacing the Vicuna-7B used in LLavA with Qwen2-7B in our setting. We hypothesize two potential reasons:
>
> - Qwen2-7B is not pretrained with vision-language alignment, unlike models such as LLaVA (based on Vicuna), which may limit its ability to process cross-modal input effectively.
>
> - Qwen2-7B has a significantly larger vocabulary than Vicuna, which may reduce fine-tuning efficiency or generalization under constrained supervision, particularly in specialized domains like ECG interpretation.
>
> In future work, we plan to explore other vision-language pretrained LLMs as decoders and study their impact on GEM’s performance across tasks.
>
> ***Table 2: Experiment with Qwen2-7B decoders***
>
> | Model            | PTB_XL(AUC) | PTB_XL(F1) | PTB_XL(HL) | CODE-15%(AUC) | CODE-15%(F1) | CODE-15%(HL) | CPSC2018(AUC) | CPSC2018(F1) | CPSC2018(HL) | CSN Accuracy | G12EC Accuracy |
> |------------------|------------|-----------|------------|---------------|-------------|--------------|----------------|--------------|---------------|---------------|----------------|
> | GEM - SFT LLavA  | 81.8       | 73.6      | 11.6       | 90.5          | 84.8        | 5.1          | 74.1           | 52.0         | 9.0           | 92.6          | 81.8           |
> | GEM - Qwen2-7B   | 80.8       | 71.9      | 12.4       | 89.3          | 83.1        | 5.2          | 75.0           | 52.8         | 9.1           | 91.1          | 81.1           |

---

> > ### Comment · Reviewer_W2KZ · 2025-08-06
> >
> > Thank you for the rebuttal. I really appreciate the work put into the rebuttal, and I appreciate the quality of the paper, and the depth of experiments performed. Based on a thorough review of the paper again, including evaluating the novelty embodied in the paper, and the depth of the experiments performed, I believe a rating of 5 is suitable for the paper. Thus, I am going to retain the original score. I applaud the authors for their systematic rebuttal and the overall quality of the paper.

---

### Official Review · Reviewer_jRz5 · 2025-07-03

**Clarity:** 3
**Significance:** 3
**Originality:** 3
**Rating:** 4
**Confidence:** 3

**Summary:**

This paper introduces GEM, designed to provide grounded, clinician-aligned interpretations of electrocardiograms (ECGs) by synergistically using ECG time series data, 12-lead ECG images, and text. The work makes several notable contributions, including a novel data generation pipeline, a new high-granularity dataset, and a clinically-oriented evaluation benchmark.

**Questions:**

1. Given that LLMs can produce factual inaccuracies, what specific quality control measures, if any, were implemented to validate the clinical accuracy of the generated interpretations? Were any samples reviewed by human experts?
2. The proposed task is evaluated using GPT-4o, which introduces potential for self-reinforcement and stylistic bias. Can the authors comment on this limitation and the steps taken to ensure the evaluation is as objective as possible?
3. The data generation method relies on GPT-4o. Could the authors provide an estimate of the computational cost and resources required to generate the 30,000-sample dataset?
4. The cardiologist evaluation in Appendix A.4 highlights several cases where expert opinions differed from the interpretations generated by GEM or GPT-4o. How do the authors interpret these disagreements?
5. Can the authors use candle plots to comprehensively reveal the stability of grounded ECG understanding for different models?

**Ethical Concerns:**

["NO or VERY MINOR ethics concerns only"]

**Final Justification:**

The strengths outweighs weaknesses, so I maintain the positive rating. However, I do believe the current manuscript can be improved regarding GPT judgement's biases, therefore I do not raise the score.

**Limitations:**

1. The primary limitation is the reliance on GPT-4o to generate the ground-truth labels for the ECG-Grounding dataset. As seen in the cardiologist evaluation, GPT-4o is not an infallible medical expert and can produce interpretations that are subtly or overtly incorrect.

2. The use of GPT-4o as the evaluator for the Grounded ECG Understanding benchmark introduces potential biases. The metric may inadvertently reward stylistic similarity to GPT-4o's outputs over pure clinical accuracy. It could be beneficial if the authors could consider metrics that supports the trustworthiness of the LLM judgement.

3. As with any clinical AI tool, there are potential risks. The primary risk is clinical error stemming from the model's recommendations, which could be compounded if the model was trained on flawed synthetic data. Over-reliance on the tool could also lead to "automation bias," where clinicians may accept an incorrect AI suggestion. Therefore more analysis on reliability considerations could benefit the paper convincing the readers.

**Quality:**

3

**Strengths And Weaknesses:**

**Strengths**:
- The paper tackles a critical problem in medical AI: the lack of "grounded understanding" in automated ECG interpretation. Medical AI requires grounded reasoning to truly help clinician make informed decisions.
- The proposed pipeline of creating grounded reasoning data could inspire research in scaling up medical AI. The release of the data could be of interest to the medical AI community.

**Weaknesses**:

- GPT-4o is not a board-certified cardiologist, and prone to hallucination and factual inaccuracies, especially in a high-stakes domain like medicine. The dataset, therefore, may contain subtle (or significant) errors, which are then propagated to the trained GEM model. The paper does not mention any quality control procedure on the created data.

- Using GPT-4o to score the model's output is problematic, as "LLM-as-a-judge" approach can be both noisy and biased. The evaluation may reward outputs that are stylistically similar to GPT-4o's own generation style rather than those that are most clinically accurate. This introduces a potential for bias and reduces the objectivity of the results for the new benchmark.

- Since the data generation pipeline largely depends on GPT-4o. This raises concerns about the cost, scalability, and reproducibility.

---

> ### Author Rebuttal · Authors · 2025-07-30
>
> Thank you very much for your constructive comments! We are delighted to see that you found the significance of the problem we studied and the value of our proposed methods. We provide detailed responses to your concerns as follows.
>
> >### **Human Evaluation Results**
>
> To fully address your concerns, we conduct a comprehensive quantitative human evaluation covering three sources of outputs: GPT-4o generated data, Deepseek-R1 generated data, and GEM generated interpretations.
>
> In total, 400 ECG-Grounding data (200 from GPT-4o and 200 from Deepseek-R1) and 200 GEM's interpretation are independently reviewed by eight board-certified cardiologists, using seven predefined clinical criteria designed to assess both reliability and usefulness. This unified evaluation protocol allows us to (1) verify the quality of GPT-4o generated training data, (2) test the effectiveness of open-source substitutes, and (3) validate the clinical utility of the GEM model.
>
> We report the evaluation results in Table 1 below and provide further discussion in the following responses.
>
> ***Table 1: Human Evaluation on GPT-4o/Deepseek-R1 Generated Training Data and GEM's Interpretation***
>
> **Reliability metrics**
> |                | Analytical Relevance | Analytical Accuracy  | Analytical Completeness  |
> |----------------|----------------------|----------------------|--------------------------|
> | **GPT-4o**     | 4.7/5                | 4.6/5                | 4.7/5                    |
> | **Deepseek-R1**| 4.8/5                | 4.7/5                | 4.9/5                    |
> | **GEM**        | 4.6/5                | 4.4/5                | 4.6/5                    |
>
> **Usefulness metrics**
> |                | Reasoning Quality | Findings Novelty | Clinical Value | Overall Satisfaction |
> |----------------|-------------------|------------------|----------------|----------------------|
> | **GPT-4o**     | 4.7/5             | 4.4/5            | 4.7/5          | 4.5/5                |
> | **Deepseek-R1**| 4.8/5             | 4.5/5            | 4.6/5          | 4.7/5                |
> | **GEM**        | 4.6/5             | 3.9/5            | 4.3/5          | 4.4/5                |
>
> ```
> Scoring criteria (Note: Due to space limitations, we only present scores 4–5 here. The full scoring criteria (including 1–3) will be provided in the updated version):
>
> Analytical Relevance: Do the model’s analyses closely support the diagnosis, and is there corresponding ECG evidence?
> 5: Every analysis point is highly relevant to the diagnosis, with clear supporting evidence.
> 4: Most analyses are strongly relevant, with minor insufficiencies.
>
> Analytical Accuracy: Are there any medical factual errors in the model's output?
> 5: Completely accurate
> 4: Mostly accurate
>
> Analytical Completeness: Does the model comprehensively discuss key ECG components relevant to the diagnosis, including rhythm, intervals, and waveforms?
> 5: All relevant ECG features (rhythm, PR, QRS, ST, T waves, intervals, etc.) are accurately discussed.
> 4: Most key ECG features are covered, with minor omissions.
>
> Reasoning Quality: Does the model provide a clear, evidence-based reasoning process similar to that of a clinician, logically deriving the diagnosis from ECG features?
> 5: Clear and coherent reasoning structure, explaining each step from ECG to diagnosis causally.
> 4: Overall reasonable reasoning, but some steps lack detail.
>
> Findings Novelty: Does the model provide insights or findings not noticed by the clinician?
> 5: Important new diagnoses or findings.
> 4: Novel and somewhat insightful content.
> 3: Some new findings, but of limited value.
>
> Clinical Value: Does the model output help in clinical decision-making?
> 5: Direct and significant support for clinical judgment; content is clear and reliable.
> 4: Most content is helpful and practically useful.
>
> Overall Satisfaction: Subjective rating of the overall quality of this analysis.
> 5: Very satisfied.
> 4: Satisfied
> ```
>
> >### **[Weakness 1, Question 1, Limitation 1, Weakness 3, Question 3, Limitation 3] Concerns on GPT-4o generated training data**
>
> - **Data quality control**
>
>     Thank you for raising this important point. We fully agree that quality control is essential when using LLM-generated data in clinical applications. In our data construction pipeline, we incorporated human expert review as a key quality control measure.
>
>     The quantitative human expert evaluation results in Table 1 shows that GPT-4o consistently achieves high scores across both reliability and clinical usefulness, with particularly strong performance in analytical completeness and reasoning quality. These results demonstrate that, with our data generation pipeline, GPT-4o is capable of generating high-quality ECG interpretations that are both clinically reliable and practically valuable.
>
>
> - **GPT-4o cost, scalability, and reproducibility**
>
>     In our implementation, generating each sample required approximately 6,000 input tokens and 300 output tokens, resulting in a total cost of roughly USD $3,000 using the OpenAI GPT-4o API for the 300,000 samples in ECG-Grounding dataset.
>
>     We recognize that GPT-4o, while powerful, is not open-sourced and introduces additional cost and access constraints, which may affect scalability and reproducibility.
>
> 	To address these concerns, we provide two solutions:
>
>      - First, given the effectiveness of our method, we have scaled up the data generation to the full MIMIC-IV-ECG dataset, which includes 784,680 ECG records from 160,597 patients. These GPT-4o generated grounding data are publicly released and intended to benefit the community as a high-quality and large-scaled resource.
>
>     - Second, to support researchers who wish to apply our method to their own datasets, we test an open-source substitute model, the latest Deepseek-R1. As shown in Table 1, Deepseek-R1 also shown strong capability of generating clinically acceptable, high-quality ECG interpretations with our methods. This demonstrates that our method is adaptable to alternative LLM backbones and remains applicable in settings without commercial API access.
>
>     We will include these results and provide a discussion of the limitations you mentioned in the revised version of the paper.
>
> >### **[Weakness 2, Question 2, Limitation 2] “LLM-as-a-judge” may introduce potential bias**
>
> We understand the reviewer’s concern regarding potential stylistic bias and self-reinforcement in using GPT-4o as the evaluation model.
>
> To mitigate these risks and ensure objective assessment, we carefully designed the evaluation prompt to include quantitative scoring criteria and mandatory justification for each score. This structured format helps constrain GPT-4o's evaluation behavior and reduces susceptibility to superficial stylistic similarity.
>
> To further mitigate potential bias, we plan to include a comparative analysis using multiple independent LLMs (e.g., Deepseek-R1, LLaMA) to evaluate the same test outputs in our updated manuscript. This will enable a more comprehensive assessment of the model outputs and help reduce the risk of bias introduced by relying on a single model for judgment.
>
> In addition, as in Table 1, our human evaluation on GEM generated interpretations demonstrate that GEM consistently achieves high scores across both reliability and usefulness dimensions, with most metrics rated above 4 out of 5.
>
> These findings indicate that GEM is capable of producing clinically meaningful and accurate interpretations that align well with cardiologists’ expectations. Also, the human evaluations on GEM are consistent with GPT-4o’s assessment, and support GEM’s potential as a trustworthy assistant for real-world cardiology applications.
>
> We will add the results and discussions in our revised paper.
>
> >### **[Question 4] Disagreements between cardiologists and models**
>
> Thank you for this thoughtful question. We interpret the observed disagreements between cardiologists and model outputs as a reflection of the inherent variability in clinical ECG interpretation.
>
> Overall, the diagnostic conclusions are often aligned between models and human experts. However, discrepancies may arise in finer-grained feature judgments, such as waveform morphology or interval thresholds. This is not unexpected, it mirrors real-world clinical practice, where interpretation differences at the detail level are common even among experienced cardiologists.
>
> Importantly, GPT-4o generated explanations are conditioned on the ground-truth diagnosis from the original clinical reports, which represent a physician’s interpretation at the time of care. As a result, GPT-4o offers one valid expert perspective, while the reviewing cardiologist may offer another. This is consistent with known inter-observer variability, particularly in morphology-sensitive cases.
>
> Rather than treating these inconsistencies as errors, we believe they represent valuable opportunities. They highlight cases where model reasoning could be refined with human feedback, and they underscore the importance of modeling expert disagreement to better capture clinical uncertainty. In future work, we plan to explore multi-expert consensus frameworks, inspired by real-world clinical workflows for handling disagreements, such as multi-physician case reviews, to further improve model reliability and alignment with clinical standards.
>
>
> >### **[Question 5] Candle plot for different models**
>
> Thank you for the insightful suggestion. While we are unable to include figures during the response period, we plan to incorporate candle plots in the revised paper to illustrate the distribution and consistency of model performance, particularly under evaluation by multiple independent LLMs as previously discussed.

---

### Comment · Area_Chair_MNVY · 2025-08-04
**Less than 3 days remaining**

This is a reminder that the author-reviewer discussion period will end soon. Please take time to read the other reviews and engage further in the discussion. The authors have put significant effort into their response, and continued discussion is important for ensuring a thorough and high-quality review process.

---

### Decision · Program_Chairs · 2025-09-17

**Decision:**

Accept (poster)

**Comment:**

This paper introduces GEM, a multimodal LLM for grounded ECG interpretation that integrates ECG time series, images, and text. Key contributions include a dual-encoder framework, a knowledge-guided data generation pipeline producing ECG-Grounding, and a new Grounded ECG Understanding benchmark. The work is well-motivated, reproducible, and supported by extensive evaluation, including cardiologist review. The main concerns were dependence on GPT-4o for data generation/evaluation, limited ablations, and lack of discussion on generalizability. The rebuttal convincingly addressed these through human expert validation, substitution experiments with DeepSeek-R1, cost/latency reporting, spirogram generalization, and additional failure case analysis. Reviewers converged on acceptance after the discussion, with some residual caution regarding LLM reliance.

The paper offers a substantial and well-validated contribution in multimodal medical AI, though the dependency on GPT-4o and limited end-to-end fine-tuning experiments temper its suitability for spotlight/oral.